# Inherent biomechanical traits enable infective filariae to disseminate through collecting lymphatic vessels

Witold W. Kilarski [1,2,6], Coralie Martin[3], Marco Pisano[2], Odile Bain[3], Simon A Babayan [4] & Melody A. Swartz[1,2,5,6]

Filariases are diseases caused by arthropod-borne filaria nematodes. The related pathologies depend on the location of the infective larvae when their migration, the asymptomatic and least studied phase of the disease, comes to an end. To determine factors assisting in filariae dissemination, we image *Litomosoides sigmodontis* infective larvae during their escape from the skin. Burrowing through the dermis filariae exclusively enter pre-collecting lymphatics by mechanical disruption of their wall. Once inside collectors, their rapid and unidirectional movement towards the lymph node is supported by the morphology of lymphatic valves. In a microfluidic maze mimicking lymphatic vessels, filariae follow the direction of the flow, the first biomechanical factor capable of helminth guidance within the host. Finally, non-infective nematodes that rely on universal morpho-physiological cues alone also migrate through the dermis, and break in lymphatics, indicating that the ability to spread by the lymphatic route is an ancestral trait rather than acquired parasitic adaptation.

[1] Institute for Molecular Engineering, University of Chicago, Chicago, IL 60637, USA. [2] Institute of Bioengineering and Swiss Institute for Experimental Cancer Research (ISREC), École Polytechnique Fédérale de Lausanne (EPFL), Lausanne 1015, Switzerland. [3] UMR7245, MCAM, Museum national d'Histoire naturelle, Paris 75005, France. [4] Institute of Biodiversity, Animal Health & Comparative Medicine, University of Glasgow, Glasgow G12 8QQ, UK. [5] Ben May Department for Cancer Research, University of Chicago, Chicago, IL 60637, USA. [6]These authors jointly supervised their work: Witold W. Kilarski, Melody A. Swartz. Deceased: Odile Bain. Correspondence and requests for materials should be addressed to W.W.K. (email: wkilarski@uchicago.edu) or to M.A.S. (email: melodyswartz@uchicago.edu)

The filariases, parasitic diseases of soft tissues, are caused by filaria nematodes that are transmitted between vertebrate hosts by blood-feeding arthropods[1]. Disease-related pathologies are induced by the death of reproductive adult filariae or their microfilaria larvae, but the specific signs and symptoms are determined primarily by the anatomical sites occupied by the parasites. For instance, out of 120 million people infected with lymphatic filariae, 30% will develop tissue swelling associated with the death of adult filariae and subsequent chronic obstruction of lymph flow[2]. The reproductive filaria worms can dwell in collecting lymphatic vessels for 15 years, and despite impairment to lymphatic contractility and the valve function, their presence leads to lymphangiectasis, the enlargement of collecting vessels, and markedly enhanced lymph flow, the cardinal signs of lymphatic filariasis[3,4]. Only when the infection is untreated, the repeated death of adult filariae and recurrent opportunistic bacterial and fungal infections eventually drive chronic lymphatic obstruction and lymphedema. Depending on the exact location of adult worms, the lymphedema might progress to hydrocele or limb elephantiasis[1,4].

The specific dwelling site of reproductive filariae is determined by the migratory infective larvae (L3) during the short and asymptomatic phase of the infection. Filaria species isolate these distinct reproductive habitats to facilitate intraspecies mating encounters while minimizing interspecies competition[5,6]. For example, larvae of geographically overlapping species of human filariae can either establish subcutaneous nodules (*Onchocerca volvulus*), migrate to the pleural and peritoneal cavities (*Mansonella perstans*) or pulmonary arteries (zoonotic *Dirofilaria immitis*)[4,7,8]. Dirofilaria-related *Loa loa* continue to wander in subcutaneous tissue even after their final molt, while phylogenetically close lymphatic filariae, *Wuchereria bancrofti*, and *Brugia malayi*[9], reside in collecting lymphatics and subcapsular sinuses of lymph nodes (LN) but located, respectively, in the thigh and abdomen, and below the knees[1].

Despite its etiological value, no mechanism has been proposed to explain the homing of filaria larvae to their specific destinations within a host. In fact, the entire physiology of the initial stage of filaria infection remains mostly presumed[10–12], and only briefly mentioned in modern textbooks[1,4,7,8,13,14]. Even the requirement of lymphatic vessels for migration of *Litomosoides sigmodontis*, a commonly used rodent model of nonlymphatic filariasis, remains speculative[15,16]. These earliest migratory events in filarial infection are challenging to study due in part to technical shortcomings of genetic manipulations of helminths that impede the use of standard imaging techniques in animals[17]. Consequently, a handful of studies that investigated the post-infection fate of filariae were limited to observations of larval migration in excised tissue[18], used inflammation as an indirect readout[17], or were performed entirely in vitro[19]. As a result, the effect of tissue migrating nematodes on the condition of local lymphatics and in consequence, the skin immunity, is poorly understood.

Here, we first show that filariae of *L. sigmodontis*, require functional lymphatic vasculature to exit the skin and eventually reach the pleural cavity, where they establish the patent infection (Supplementary Note 1). It is, therefore, a suitable organism to explore the mechanism of lymphatic invasion by filariae. A method of fluorescent labeling of live filariae, combined with adapted imaging techniques—intravital epifluorescence, confocal optical sectioning of whole-mounted skin[20–22], and near-infrared macroscopy—allow us to analyze processes that have not previously been visualized in a live animal. This includes migration of infective larvae within the skin, invasion of precollecting lymphatic vessels, and subsequent intralymphatic escape to the LN. The bioengineering application of a custom-built microfluidic maze that mimick lymphatic collectors permits the correlation between the directionality of filaria migration and the fluid current, the universal biomechanical factor always present in living tissues[23]. Finally, we show that even nonparasitic nematodes, such as bacterivores and plant parasites, can migrate in the dermis and forcibly enter lymphatic precollecting vessels. This suggests that the ability of filariae to invade lymphatics and colonize the host is an ancestral or symplesiomorphic trait shared at least by filariae and free-living nematodes, that is, the majority of zooparasitic and terrestrial nematodes[24].

## Results

**L. sigmodontis requires functional collecting lymphatics.** To determine whether *L. sigmodontis* exclusively uses lymphatics to reach the pleural cavity (Supplementary Note 2), we used primary (Chy3) and secondary (induced) mouse lymphedema models. Chy3 mice have reduced lymphatic vessel density in the skin and occasionally develop occlusion of terminal hind–limb lymphatic collector[25,26]. Mice with induced secondary lymphedema have surgically excised terminal lymphatic collectors within the tail. In both these models, a complete blockade of the lymphatic route is manifested with persistent edema within affected appendages.

In the control mice, 23 (SD 11%) (Fig. 1a) to 21 (SD 10%) (Fig. 1b) of larvae inoculated subcutaneously in the shank region of the hind leg or the tail were recovered from the pleural cavity (Fig. 1a, b), which is in agreement with previous publications[16,27]. However, likely due to variability within the phenotype, 7 (SD 6%) of filariae were still able to reach the pleural cavity in Chy3 mice (Fig. 1a, nonparametric, one-tailed Mann–Whitney $t$ test: $p = 0.0003$, the effect size 2.7), showing strong correlation but not unequivocal dependence of *L. sigmodontis* on the lymphatic route. In the secondary lymphedema model, none of the subcutaneously injected filariae could be later found in the pleural cavity (Fig. 1b, $p = 0.00021$, Fisher exact test 3 to 2 table of 5-0, 8-0, 0-4). The fact that no filariae could exit the tail skin when lymphatic drainage was blocked entirely indicated that lymphatics are the only migratory routes used by *L. sigmodontis* filaria to reach its extracutaneous habitat and therefore established patent infection.

**Fluorescently labeled filariae retain their infectiveness.** For in vivo imaging of L3 larvae, we fluorescently stained them with amine-reactive isothiocyanate derivatives of rhodamine (TRITC, Fig. 1c–g) or near-infrared dye VivoTag 680 XL NHS ester (Supplementary Fig. 1). Fluorescent labeling was uniform between filaria larvae, highlighting their anatomical structures, which later allowed determination of the head-to-tail orientation of larvae in fixed tissues (Fig. 1c, Supplementary Note 3). Next, we tested if chemical labeling of filariae had an immediate detrimental effect on larvae migration in vivo.

For a whole-body imaging, VivoTag 680 XL-labeled larvae were injected subcutaneously in the dorsolateral lumbar area and imaged over time (Supplementary Fig. 1, Supplementary Movie 1). In a single experiment, we calculated filariae spread from the injection site and the efficacy of their systemic escape (Supplementary Fig. 1c, Supplementary Movie 1). Out of the 39 inoculated larvae, 29 escaped the injection site, a comparable efficiency to the reported subcutaneous (over 70%) spread of related *Brugia pahangi*[28]. After 20 h, 9 of the 29 migrating filariae had escaped the skin, which is within the range of the infection efficiency shown in Fig. 1a, b (22 (SD 10%)), but also previous reports (15–25%)[16,27]. Labeled filariae could survive and migrate in the skin for at least 18 h (Supplementary Fig. 1c). This experiment showed that fluorescent labeling had no immediate

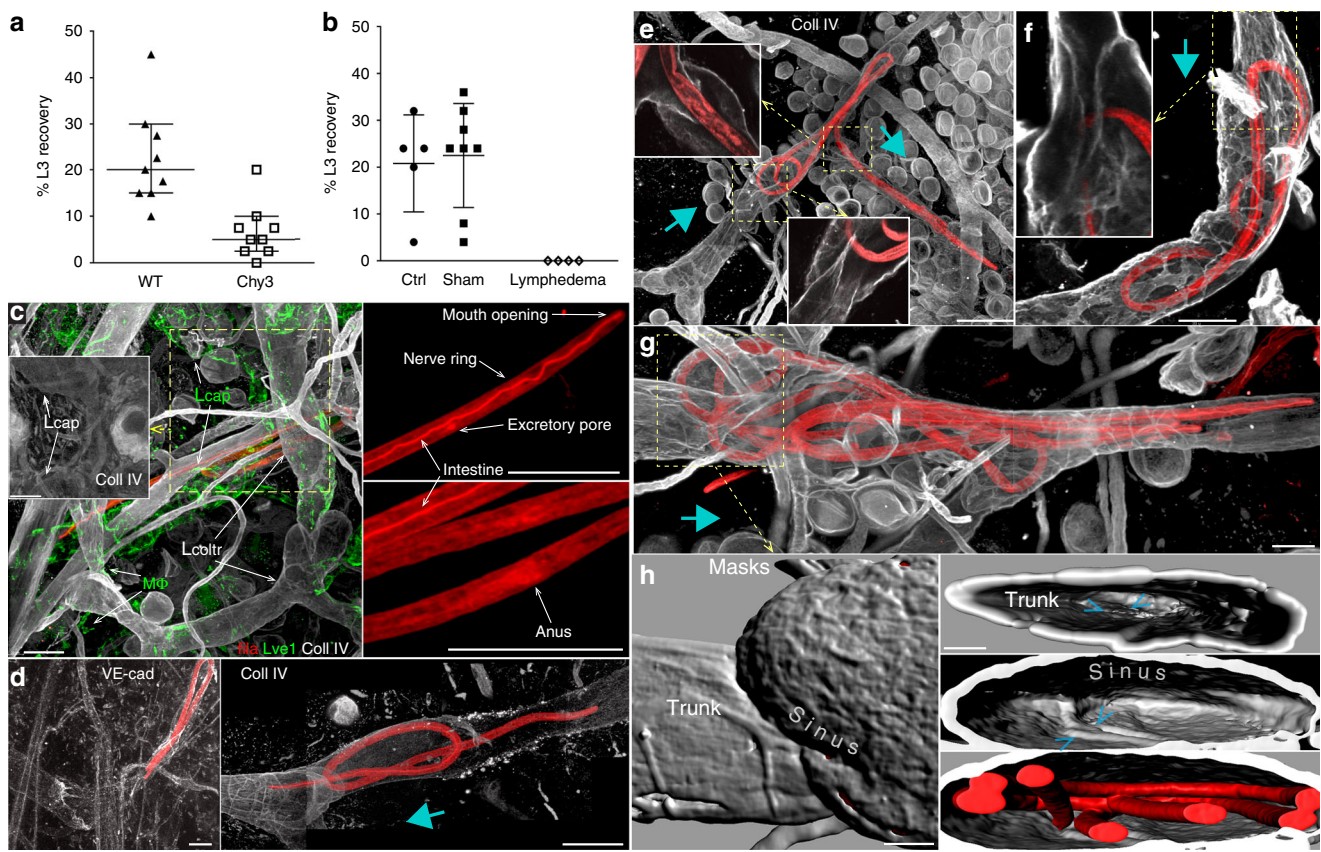

**Fig. 1** Lymphatic valves direct migration of filaria larvae toward the lymph node. **a** Model of primary lymphedema. Success rate for *L. sigmodontis* filariae migration to the pleural cavity (as % inoculated 10 days earlier) after inoculation in lymphedematous hind paw of Chy3 mice. Totally, 40 filariae were injected in each of 9 control and Chy3 mice ($p = 0.0003$, one-tailed Mann–Whitney $U$ test). **b** Model of secondary lymphedema. Success rate for *L. sigmodontis* filariae migration to the pleural cavity after inoculation into lymphedematous tails. Totally, 40 filariae were injected in 6 (control), 8 (sham), and 4 (lymphedema; $p = 0.00021$, two-tailed Fisher exact test (3 × 2 table). For **a**, **b**, individual results (with means and standard deviations) of two pooled experiments are shown. Source data are provided as a Source Data file. **c–h** Maximum intensity projections (MIP) of immunostained whole-mounted dorsal ear dermis with migrating filariae. **c** Left, TRITC-labeled filariae (fila) outside the Lyve1 (Lve1)-stained lymphatics capillaries ($L_{cap}$) and collagen IV (Coll IV)-stained collectors ($L_{coltr}$). Mφ-macrophages. Inset. One micrometre optical section shows basement membrane of capillary lymphatic. Right, fluorescent labeling of a larva revealed the presence of anatomical organs and allowed identification of its head-to-tail orientation (Supplementary Note 3).
**d–h** Intravascular larvae were found only in lymphatic precollectors. **d** Left, In contrast to blood vessels, lymphatic precollectors are weakly stained with VE-cadherin (VE-cad). **d** Right, Strong signal from collagen IV revealed the characteristic morphology of lymphatic collectors. The larva that is crossing the valve is located at the trunk-side of the lymphatic valve. **e–h** Valves of lymphatic collectors prevent afferent filariae migration. **e**, **f** Insets. Directionality of lymphatic valves on 1.8 μm optical cross-sections. **g** Three larvae coiled at the post-valve sinus. **h** The structure of the valve. Surface masks created from the MIP image shown in (**g**). Migrating larvae have a higher probability of localizing the inlet of the lymphatic trunk than the outlet of the lymphatic sinus because the valve opening occupies a proportionally larger area of the smaller prevalve lymphatic trunk. Blue arrows in (**d–g**) indicate the original directionality of lymphatic drainage. Blue arrowheads in (**h**, right) point to the edges of the valve in the trunk (top) and the sinus (middle). Bars = 50 μm

detrimental effects on the in vivo migration and the systemic dissemination of infective larvae.

For intravital and confocal imaging of filaria in live and whole-mount skin preparations, we inoculated TRITC-labeled L3 filariae in the dorsal dermis of a mouse ear, an optimal site for epifluorescence imaging[20]. However, the ear skin did not facilitate efficient larvae inoculation, as approximately 80–100% of larvae remained in the intradermal injection pocket after inoculation (Supplementary Movie 2, Supplementary Note 4). The in-pocket entrapment could be reproduced in vitro by allowing the larvae to form a void in the matrix during gel polymerization (Supplementary Movie 3); without a surrounding physical matrix, the larvae could not propel itself directionally. As the effects of filariae entrapment within postinjection dermal voids was a treatment-related artifact it was not further analyzed as a part of the filariae migratory process. Nevertheless, a fraction of inoculated larvae

escaped the injection pocket and migrated into the dermis where we found total 23 larvae located within precollecting lymphatics (eight are shown in Fig. 1d–g), but never within a blood vessel.

Staining for Lyve1 identified skin capillary lymphatics but was scattered or completely absent in larger lymphatic precollectors (Fig. 1c, left). The alternative staining for VE-cadherin, a pan-endothelial marker, produced a weak signal in the lymphatic endothelium and could not reliably expose the morphology of lymphatics (Fig. 1d, left). In contrast, a strong signal from the basement membrane (BM) stained for collagen IV revealed most of the anatomical details of lymphatic precollectors (Fig. 1d, right), and additionally allowed the morphology-based determination of their original directionality of the lymph drainage (Fig. 1d–g; Supplementary Note 5). Weak but uniform staining for BM around capillary lymphatics could be identified on individual optical sections (Fig. 1c, left).

**Valves direct filaria migration within lymphatics**. Here, we tested if the same lymphatic valve architecture that maintains a passive unidirectional lymph flow could independently favor passage of actively migrating L3 larvae toward the draining LN. The direct effect of lymphatic architecture on larvae migration can be separated from the lymphatic-controlled lymph flow in anesthetized animals, where physiological lymph currents are the lowest[29–31] (Supplementary Note 6). Three hours after intradermal inoculation, that is, after the injection-driven edema and lymph flow plateau, filariae that entered lymphatics were distributed asymmetrically around lymphatic valves. Out of 11 larvae (out of 23 located within lymphatics) found in the proximity of valves, only two filariae were located at the trunk (inlet) afferent side of the valve (one is shown in Fig. 1d, Right), while 9 filariae were found at the sinus-(outlet) efferent side of the valves (six larvae are shown in Fig. 1e, f; $p = 0.0654$; the two-tailed signed test for 2 vs. 9 filariae). As our hypothesis postulated a presence of a universal to nematodes mechanism that directs the migration of filariae within lymphatics, we pooled the observations of filariae ($n_{filariae} = 23$: $n^+ = 9$, $n^- = 2$, $n_{null} = 12$) and, discussed later, soil-derived nematodes ($n_{nematodes} = 3$: $n^+ = 2$, $n^- = 0$, $n_{null} = 1$). As a result, out of 26 nematodes found within lymphatics, 11 nematodes were found at the sinus, and 2 at the trunk side of the lymphatic valves ($p = 0.0225$, the two-tailed sign test for 2 vs. 11 nematodes).

The prolonged time that filariae spent at the sinus side during in-lymphatic migration can be explained by the varying success rate of crossing the same passage from afferent (trunk) or efferent (sinus) side of the asymmetrical valve. Thus, filaria has a lower probability of locating valve outlet from the sinus side because in the sinus the outlet occupies a proportionally smaller fragment of the wall as compared to the trunk side of the valve (Fig. 1g, h). However, once larvae select the downstream direction, an extended wall of postvalve sinuses should provide larger resistance surface than the wall of the trunk allowing the filaria tail to push off against the vessel wall and propel its migration. Together, the architecture of postvalve sinuses might act as a functional valve assuring unidirectional migration of parasites.

**Filariae migrate differently in dermis and lymphatics**. Using a modified intravital immune epifluorescence technique[20], we characterized the migration pattern of filariae in the dermis and within lymphatic collectors (Supplementary Note 7). Most of the intradermally injected filariae remained in the skin inoculation pocket with only a few larvae entering the dermis. Even then, some of the dermally migrating larvae moved vertically toward the surface of surgically exposed skin and left the dermis into the overlaid media (Supplementary Movie 4 (the third appearing larva) and Supplementary Movie 5; Supplementary Note 8). Only a fraction of inoculated filariae migrated into the imaging area of the surgically exposed and prestained dorsal ear dermis (Supplementary Movies 6–8 (the first 45 min)). The direction of instantaneous velocities of a single filaria that migrated over the long distance of the skin (Supplementary Movies 7) is shown in Fig. 2a, b. Larvae migrating in the dermis spent most of the time probing the dermis, either being inactive or making short backward and forward movements (head and tail movements). These periods of negligible migratory progress (lag phases, lags) were interleaved with rapid bursts in larval migration, during which larvae advanced at the average maximum speed (see Terms used in Supplementary methods) of $V_{derm} = 106.5$ (SD 23 µm/min) (weighted mean from four migrating larvae, the source data for all speed calculations are provided as a Source Data file). During migratory bursts, larvae made the net translocation, while migratory lags accounted for the majority of the time filariae

spent in the dermis. The biphasic migration pattern was also characteristic for a single larva that was imaged during its migration within lymphatic precollector (Supplementary Movie 9, Fig. 2c, d). Contrary to intradermal migration, filaria migration within the lymphatic vessels was unidirectional, and its maximum speed ($V_{lymph} = 731$ µm/min, three bursts combined in a single regression) was over sixfold faster than $V_{derm}$. In lymphatics, migratory bursts and interrupting lags were of comparable duration (Supplementary Note 9). Two of these lags (Lag$_1$ and Lag$_2$ in Fig. 2d) corresponded to larvae passing complex junctions with more than two valves (V$_1$ and V$_2$ in Fig. 2c, e). The larva used the structural support of the collector wall, as it could be inferred from the deformation of the BM during filaria passage (Supplementary Movie 10).

**Filaria larvae can rapidly reach the LN subcapsular sinus**. Inoculated larvae were able to reach the draining LN within 18 min after intradermal inoculation (Supplementary Fig. 3a), the shortest time needed for mouse perfusion and excision of LNs. We found a total of 15 filariae in four LNs, two of which with 8 larvae were collected immediately (18 min between larval inoculation and LN excision), and (two others with 7 larvae) collected 3 h after inoculation. These results show that infective larvae can localize and invade lymphatic vessels and reach the draining LNs within minutes after entering the skin (Supplementary Note 10).

**Lymphatic collectors are injured during entry of infective filariae**. Here, we analyzed the impact that filaria invasion has on the integrity of collecting lymphatics. Injuries to lymphatic collectors are inflicted by the granuloma triggered by the death of the adult filaria in the advanced stage of lymphatic filariasis[1], but no reports showed the effect of L3 filaria larvae on lymphatics. Infective larvae forcefully entered lymphatic vessels, resulting in the variable loss of vessels integrity, with injuries ranging from a puncture wound in the lymphatic BM (Supplementary Fig. 4a) to the complete disconnection of vessels continuity (Fig. 3a, b, Supplementary Fig. 4b). We were able to track the entire migratory path of the in-lymphatic migrating filaria from Supplementary Movie 9, up to the lymphatic entry site, where lymphatic damage occurred. Using the BM structures as guideposts, first we identified the collecting vessel used by the migrating filaria and imaged this vessel until the distal location in the dorsal dermis, where dorsal and ventral skin fragments were left intact for larvae inoculation. From this point, we imaged all branches of precollecting lymphatics (inaccessible during the intravital imaging) that could potentially be invaded by the migrating larva (Figs. 2c–e, 3b, and Supplementary Fig. 2). This led to the identification of a single, 200-µm-long lymphatic breach within one of the afferent lymphatic branches (Fig. 3b). The gap was flagged at the opposite sides of the lymphatic breach by differential staining of CD31 (PECAM-1), but not VE-cadherin in the lymphatic endothelium (Supplementary Note 11). Variable levels of CD31 at the anterior and posterior sites of the lymphatic breach reflected the post-injury remodeling changes within the surviving endothelial cells. The presence of folded BM within the breach at the afferent side of the lymphatic suggested that invading larvae mechanically pushed a fragment of the precollector relocating it towards the center of the vessel. The folding of the lymphatic wall could be caused by forceful mechanical relocation (pushing and squeezing) but not with the enzymatic digestion of the vessels, the previously suggested mechanism of lymphatic entry[13].

Filariae entirely confined within lymphatics were found only in collecting lymphatics (total of 23 larvae in 14 inoculated ears). In a single case, we found a larva penetrating two Lyve1-positive

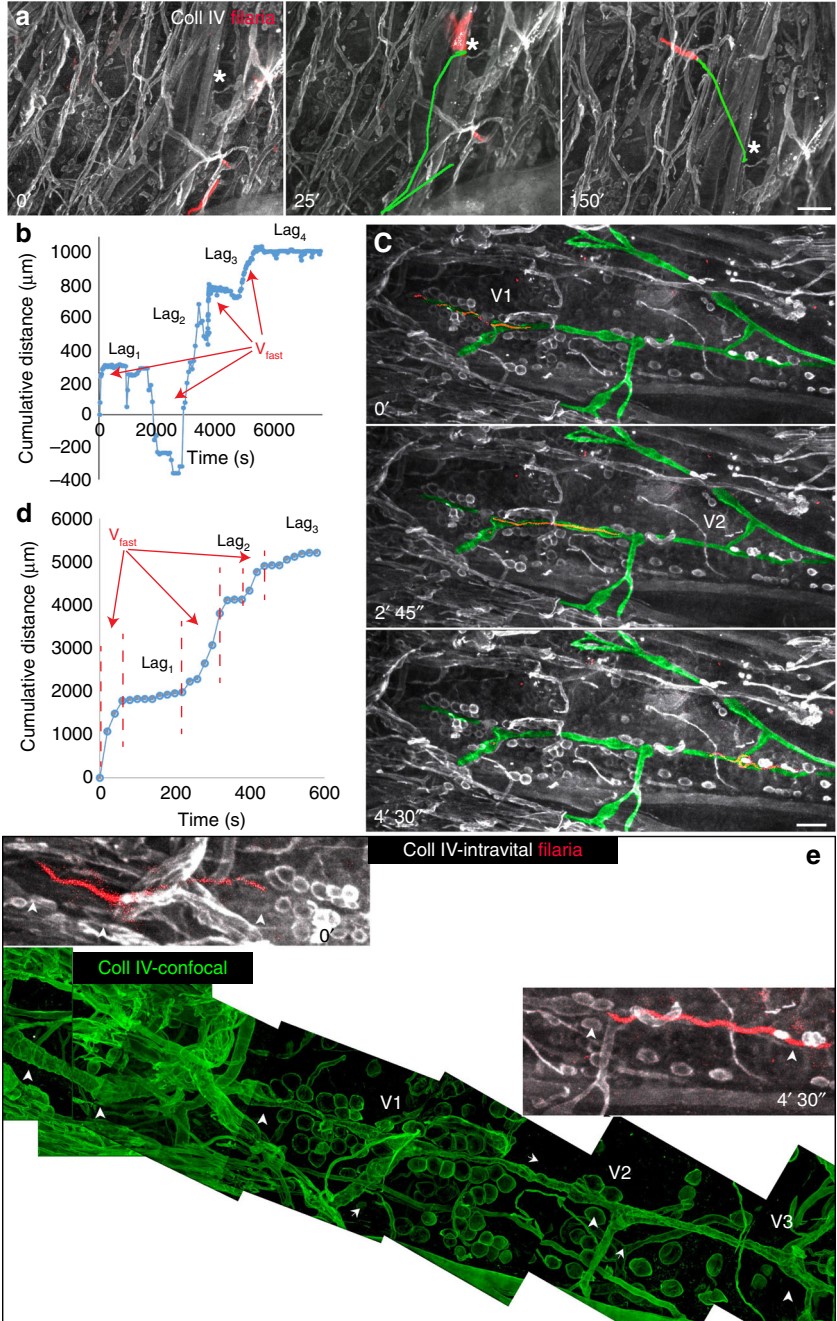

**Fig. 2** Different modes of filariae migration in dermis and within lymphatics. Intravital imaging of live-labeled filariae in the tissue stained for collagen IV (Coll IV). **a**, **b** Filaria migration in the dermis. **a** Filaria displays a discrete pattern of migration in the interstitium. Image sequence of 12-h tissue probing by larvae migrating within the dermis (Supplementary Movie 7). The periods of negligible migratory progress were interleaved with rapid bursts of migration. *, tissue reference point; green lines, migration paths from the preceding time point (shown in the middle image). **b** A graph showing the biphasic migratory pattern, and the directionality of filariae migration within the dermis interstitium. The net migration took place during migratory bursts (arrows), but larva spent most of the time in backward–forward movements (migratory lags). **c**, **d** Filaria migration in lymphatics. **c** Biphasic and unidirectional migration of filaria in lymphatics. A 13-min image sequence of the filaria larva migrating within the lymphatic collector (Supplementary Movie 9). The system of lymphatic collectors is marked by the green mask manually painted over the image. $V_{a1}$ and $V_{a2}$ mark locations of valves where a filaria paused its migration. **d** A graph showing the speed and the unidirectionality of filaria migration inside the lymphatic collector. Migratory bursts (arrows, $V_{lymph} =$ 731 μm/min) were interleaved by equal migratory lags. Source data are provided as a Source Data file. **e** Confocal imaging of the reconstructed migratory path revealed that intralymphatic migration of larva was delayed only at complex lymphatic junctions (V1–V2). Arrowheads point to all valves of the collector. White, images from the intravital imaging with the fluorescence stereomicroscope that preceded fixation and subsequent confocal imaging (maximum intensity projections, green) of the same skin fragment. Tracking of movement of a single filaria in the dermis (**b**) and in lymphatics (**d**) (source data are provided as Source data file). Scale bars 100 μm

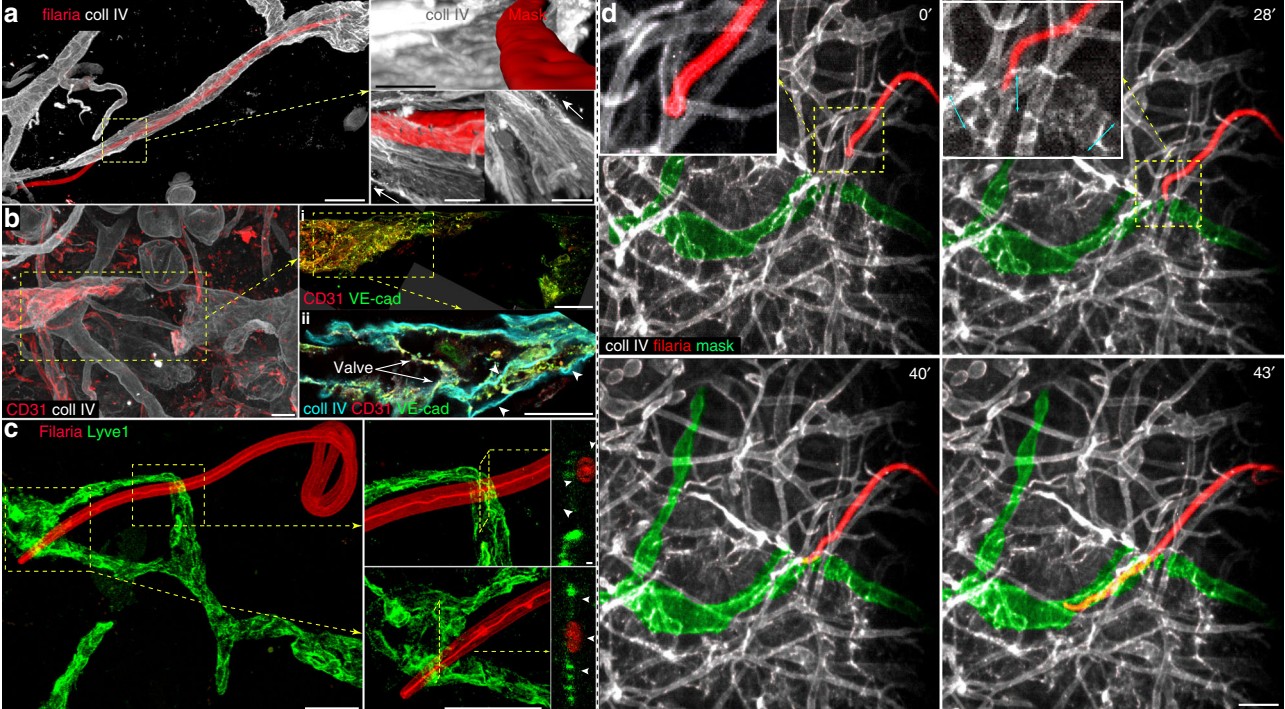

**Fig. 3** Forceful filariae entry results in lymphatic injury. The process of a larva entering into a lymphatic vessel resulted in various levels of lymphatic damage. **a** The loss of the large portion of the basement membrane (BM) and reorganization of its remnants accompanied larval entry into the collagen IV (Coll IV)-stained lymphatic collector. Left: The collecting lymphatics partially invaded by filaria. The dashed box encloses the filaria entry site magnified in the images in the right panel. Right: The damage and rearrangement of the remaining BM. Arrows point to the directional reorganization of the BM around the site of entry of the larva. **b** Filaria entry caused damage to both BM and the endothelial lining of the collector. The extrapolated site of entry (dashed box) of the filaria that was pictured migrating within lymphatic in the Supplementary Movie 9. Ten hours after the filaria crossed the collectors, the mouse was perfused-fixed, and the entire dorsal ear dermis was restained for collagen IV, and stained for CD31 (PECAM-1) and VE-cadherin. **b** (panel i) The variable presence of CD31 but not VE-cadherin (VE-cad) indicates that the remodeling of the collector took place before the fixation. Image combined from three ×100 magnification fields. **b** (panel ii) A 160-nm cross-section of the area dashed in bi revealed a presence of the additional chamber (arrowheads) composed of BM and the endothelial lining, suggesting a collapse of the opposing wall of the collector. **c** The filaria penetrated through two Lyve1-positive capillary lymphatics but did not enter their lumen. Arrowheads in 450 nm cross-sections point to the remnants of the cellular wall of the capillary lymphatic. **d** The image sequence of the filaria entry into lymphatic collecting vessel from Supplementary Movie 7. The green delineation (mask) was manually painted over the invaded lymphatic precollector. The inset in the top row shows the head of the larva while burrowing in the blood capillary (left) and lymphatic collecting vessel (right). Coll IV-collagen IV, VE-cad-VE-cadherin. Scale bars: **a** Left, 200 μm, **a** Right, 10 μm, **b** 20 μm, **c** 50 μm (inset 5 μm), and **d** 100 μm

capillary lymphatics (Fig. 3c). The larva crossed two lymphatic capillary lumens, which, in contrast to collecting vessels, did not reorient the penetrating head of the filaria (compare with Fig. 3a and Supplementary Fig. 4a, b). Capillary lymphatics are likely to be only an incidental target of infective filariae because without the solid BM support their wall cannot redirect larvae migration or resist their exit from the lymphatic lumen.

Finally, we were able to capture the moment of a larva entering into a lymphatic precollector (Supplementary Movie 8). The sequential analysis showed that the larva spent 41 min on probing the tissue before reaching a blood capillary (Fig. 3d, 0'). The larva failed to puncture the capillary (28') and continued burrowing within the tissue until it reached collecting lymphatic vessels (40'). From there, it took only 3 min to breach the lymphatic wall and enter the precollecting vessel (43', 83 min after inoculation).

**Fluid flow directs the movement of filaria larvae.** Here, we wanted to verify if filaria migration is guided by the tropism to fluid currents. Because the generation of lymph from interstitial fluid drops to minimal levels in restrained or anesthetized animals[29–31], the effect of flow on larvae migration was analyzed using a microfluidic device. The custom-designed hexagonal

layout of the microfluidic maze (Supplementary Note 12) allowed generation of velocity gradients spanning two orders of magnitude (Fig. 4a).

Comparison of larval migration in static and flow conditions showed that filariae responded to applied fluid currents by changing the direction of their movement (Fig. 4b, Supplementary Movie 11). The directionality of movement of each larva at consecutive time points ($t_i$) was expressed as the relative net displacement ($Rf_i$), a coefficient that can take a range of values from 0 for random to 1 or −1 for unidirectional migration. In the presence of flow, the (1) $\overline{Rf} = \frac{\sum_{i=1}^{n} Rf_i}{n}$) was positively correlated with the vector of applied flow ($\overline{Rf}_{\text{flow}} = 0.845$ (SD 0.31), $p < 0.0001$ for comparison to hypothetical $\overline{Rf}_{\text{h}} = 0$, one-sample $t$ tests). In contrast, in the absence of a directional flow, filariae movement was random (Supplementary Movie 12; ($\overline{Rf}_{\text{stat}} = 0.11$ (SD 58), $p < 0.63$)). The strong, 1.57, effect size of the flow indicated that it could direct 88% of randomly moving larvae toward the exit of the maze (Fig. 4d, ($\overline{Rf}_{\text{flow}}$) vs. ($\overline{Rf}_{\text{stat}}$), $p = 0.007$, Mann–Whitney $U$ test). The speeds ($V$) of filariae migrating within the maze were not affected by the flow ($V_{\text{stat}} = 452$ (SD 342 μm/min), $n = 7$ vs. $V_{\text{flow}} = 547$ (SD 464 μm/s), $n = 7$,

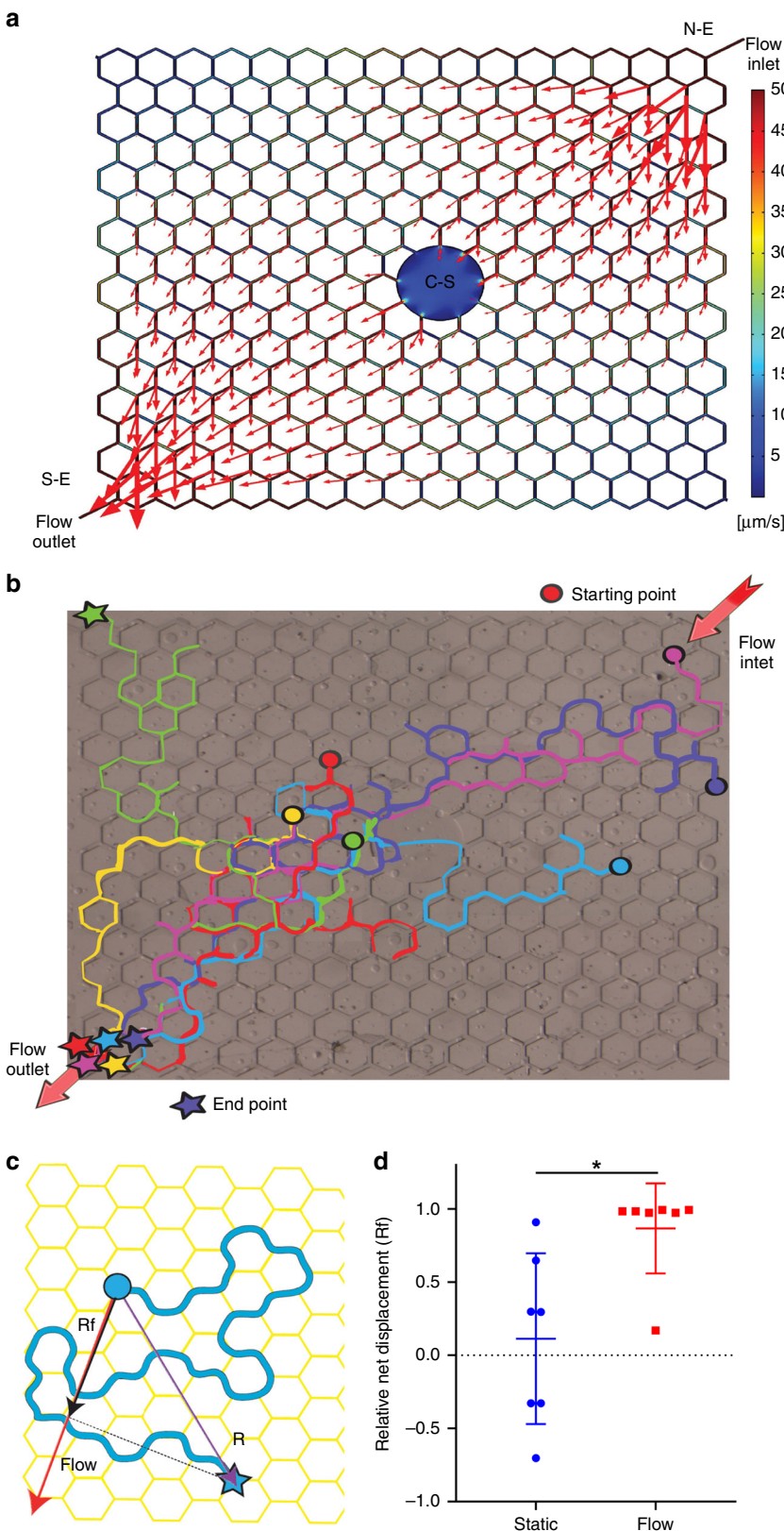

$p = 0.67$, unpaired $t$ test), indicating that that larvae actively navigated within channels and were not merely flushed by the surrounding fluid.

**Soil-derived nematodes enter lymphatic collectors.** Our results suggested that the mechanism of filaria dissemination was dependent on universal morpho-behavioral features of their larvae. Next, we wanted to verify whether these abilities are derived parasitic adaptations (filaria-specific apomorphies), or, contrary, they represent ancestral traits (plesiomorphies) that would be shared between all filaria species, but also with nematodes from filaria sister groups, that is, with their terrestrial relatives[32].

**Fig. 4** Larval migration is directed by the fluid flow current. **a** The geometry of the microfluidic maze and its flow pattern. Larvae were pipetted into the chamber through the Central Station (C-S) and allowed to spread through the chamber. Later the pump was connected to the flow inlet (N-E), and applied flow generated a gradient within the channels. The red arrows represent the flow velocity vectors, and their length represents the flow magnitude. The magnitude of the flow is also color-coded within the channels of the maze. **b** Representative data showing larval migration paths over 30 min tracking under flow conditions; each color represents a path of a distinct larva. Note that most larvae ended their migration at the flow outlet (S-E). Circles show the larvae starting position and stars denote their position at the end of the 30-min migration. **c** Schematic showing the method for calculation of the net displacement index ($R_f$, black arrow) that was calculated as the dot product of the larvae net displacement vector ($R$, purple arrow) and the vector representing the highest magnitude of flow (red arrow). **d** Relative net displacement values for the static and flow conditions, indicating that the directionality of the larval migration was dependent on flow as compared to static condition ($p = 0.007$), Mann–Whitney $U$ test, comparison of experiments with an equal number (7) of migrating filariae (Source data are provided as a Source Data file). The graph shows individual filariae $R_f$ values, their means with their standard deviations. In total, the flow experiments were performed five times and static three times (see Replication in Supplementary methods for details)

To do that we imaged dermal migratory behavior of soil-derived nematodes (Supplementary Note 13). These free-living terrestrial nematodes along with the largest group of vertebrate and plant parasites belong to the subclass *Chromadoria*, the largest group of *Nematoda*[24]. These terrestrial nematodes are characterized by the feeding behavior, which excludes the acquisition of any parasitic adaptations that would allow them to navigate throughout the vertebrate tissues or enter host lymphatics (Supplementary Note 14). However, soil-derived nematodes and zooparasitic filariae share general morphological and behavioral traits, such as, the longitudinal muscle arrangement that together with impermeable to solutes trilayered cuticle is responsible for their bending movement[32] (Supplementary Movie 13).

Similar to filaria larvae, intradermal inoculation of soil-derived nematodes resulted in their entrapment within the needle pocket of the ear skin (Supplementary Movie 14). However, the varying sizes and movement characteristics of isolated nematodes presented further technical challenges (Supplementary Note 15) that imposed modifications to the intravital imaging protocol. Instead of the surgical exposition the dorsal dermis for staining and imaging, TRITC-labeled soil-derived nematodes were imaged directly through the dorsal dermis. The effect of phototoxicity was inferred from the sudden change in the movement of nematodes and was addressed by the introduction of 5–10 min imaging pauses (Supplementary Movie 15).

Implanted terrestrial nematodes migrated within the ear dermis following a pattern similar to that of the filaria larvae; that is, multidirectional movement with prolonged periods without net advances interleaved with short migratory bursts (Supplementary Movies 16 and 17). The maximum (burst) velocity of in-dermis migration of soil-derived nematodes ($V_{der}^{nema} = 138$ (SD 56 µm/min) that was calculated for six soil-derived nematodes (the source data are provided as a Source Data file) was comparable ($p = 0.32$, unpaired $t$ test) to the velocity of filariae in-dermis migration ($V_{derm}^{filar} = 106.5$ (SD 23 µm/min), calculated for four filariae, Fig. 2a, b). The migratory pattern of a single nematode traversing a long distance (over 30%) of the dorsal ear (the second nematode that emigrated from the inoculation site in Supplementary Movies 18 and 19) resembled the in-lymphatics migration of a filaria larva (Fig. 2c, d); that is, it was unidirectional with comparable amount of time spent on the migratory advances and migratory lags (Fig. 5a). Three fast migratory bursts of that nematode ($V_{fast}^{nema} = 635$ µm/min, combined in a single regression, the source data are provided as a Source Data file) were interleaved with three lags in movement but also with four periods during which the nematode migrated at a slower velocity (132 µm/min; Supplementary Note 16). The extreme differences between two boundary velocities of the soil-derived nematodes (effect size 8.9) for $V_{fast}^{nema} = 635$ µm/min and

$V_{derm}^{nema} = 138$ (SD 56 µm/min) also resembled the effect size of lymphatics on the speed of filariae migration (effect size 27.2) for the comparison of in-lymphatic $V_{lymph}^{filar} = 731$ µm/min, $n = 1$ and in tissue $V_{derm}^{filar} = 106.5$ (SD 23 µm/min) velocities ($n = 4$).

In the confocal reconstruction of the nematode pathway, we managed to overlap the majority of the migratory route of the nematode from movies 18 and 19 with the probable lymphatic path. Within that path, we identified two complex junctions of precollectors, which similar to filaria in-lymphatic migration, could be responsible for the 2 out of 3 lags that occurred during the nematode migration (Supplementary Fig. 5).

In five imaged whole-mounted skin flaps we found 26 nematodes that migrated outside the inoculation pocket. Three of these nematodes with diameter 9.7 (SD 2.8 µm) were located entirely within collectors (two nematodes that were located at sinus sides of their valves are shown in Fig. 5b, c, e) and three others (diameter 15.4 (SD 4.1 µm) only partially crossed the wall of lymphatic collector (one is shown in Fig. 5d). Two more plant-parasites and 18 bacterivores were found outside vascular structures (diameter 19.6 (SD 9.7 µm)). The diameter of nematodes was inversely correlated with their ability to enter lymphatics (the effect size of the nematode width was 0.93; $p = 0.044$ for 6 nematodes partially and entirely within lymphatics), Mann–Whitney $U$ test; Supplementary Note 17). The diameter of the nematodes that were entirely enclosed within lymphatics (diameter 9.7 (SD 2.8 µm, $n = 3$)) was comparable to the diameter of *L. sigmodontis* L3 filariae (diameter 10.1 (SD 1.4 µm, $n = 45$, $p > 0.99$)). Contrary, L3 filariae differ in diameter from non-infective nematodes that were partially enclosed within collectors (diameter 15.4 (SD 4.1 µm, $n = 3$, $p = 0.047$, effect size 1.7)), and significantly (effect size 6.6) from nematodes found in the dermis but outside lymphatics (diameter 19.6 (SD 9.7 µm, $n = 20$, $p < 0.0001$; Kruskal–Wallis test with Dunn's correction)). This suggests that there is an optimal diameter that allows a nematode parasite to invade lymphatics. However, while both L3 filaria and smallest nematodes could fit in lymphatics capillaries (diameter 29.3 (SD 9.1 µm, $n = 24$ Lyve1-stained vessels of ear skin)) and precollectors (diameter 42.9 (SD 15.0 µm; $n = 24$ collagen IV-stained skin vessels)), only precollectors provided the conditions that permit the complete invasion of a vessel (Source data for the above comparisons are provided as the Source data file). Five hours after inoculation, we were able to localize a single (Supplementary Note 18) actively moving soil-derived nematode at the subcapsular sinus of the draining LN (Supplementary Movie 20), indicating that, similar to filaria larvae, soil-derived nematodes, could use lymphatics to escape the skin.

The fact that even terrestrial nematodes, which are not evolutionarily adapted to parasitism on vertebrates, were able to migrate within the dermis and invade lymphatics suggests that lymphatic vasculature might be a universal route of systemic

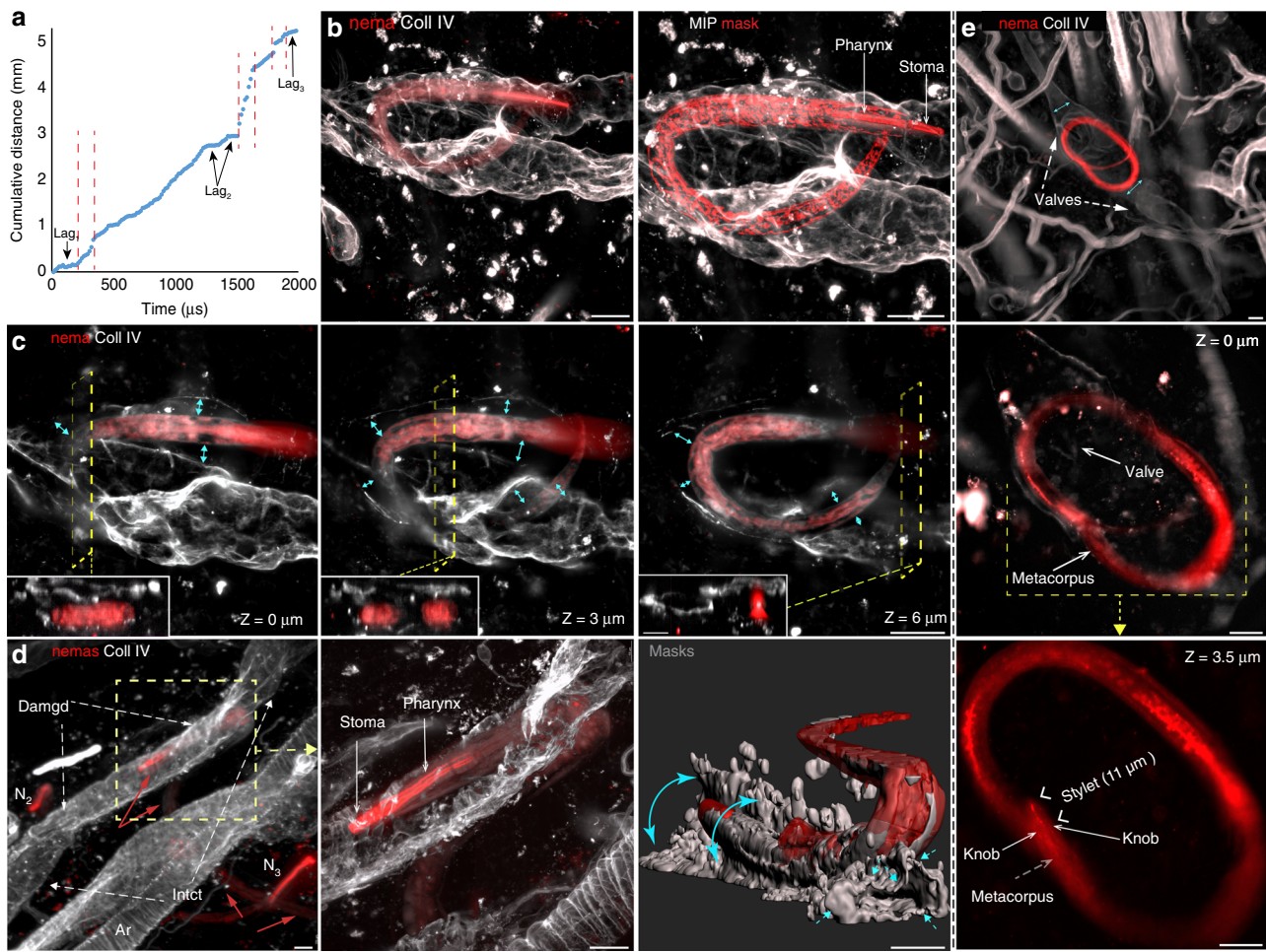

**Fig. 5** Soil nematodes migrate through the dermis and enter collecting lymphatics. Confocal images of whole-mounted skin inoculated with TRITC-labeled soil-derived nematodes (nemas) and stained for collagen IV (Coll IV). **a** Soil-derived nematode displays a discrete pattern of in-dermis migration (analysis performed on a single nematode). Graph shows the speed and unidirectionality of migrating nematode (Supplementary Movies 18 and 19) with three distinct phases: backward–forward movements (lags, no migration), slow, and burst migration ($V_{fast} = 635\ \mu m/min$, arrows). **b–d** Bacterivore nematodes within a lymphatic collector. **b**, **c** A bacterivore nematode located entirely within a collecting lymphatic vessel. **b** Left, maximum intensity projection (MIP) of the bacterivore with strongly labeled unmodified tubular stoma and pharynx[62,63]. **b** Right, MIP of collagen IV and a surface mask over the entire signal range of the nematode. Left panel **c**, 3.5 μm optical XY sections show that the nematode shown in (**b**) was enclosed by the basement membrane (BM) wall of the collector. Double arrows are drawn between the lymphatic wall and the nematode. Insets show 2-μm YZ cross-sections of the stack. **d** Damaged to lymphatic collector invaded by bacterivore nematode. **d** Left and **d** Middle, collector partially invaded by $N_{inv}$ nematode. The BM of the invaded lymphangion had a disorganized composition as compared to the BM in the surrounding segments. $N_{inv}$, bacterivore that partially invaded lymphatic collector, $N_2$ and $N_3$, two nematodes within interstitium, Ar-artery. **d** Right, A single intensity surface masks over the entire signal range of the nematode (red) and collagen IV (white). Head of the bacterivore located within the projected lumen of the collecting lymphatic. Lost wall of the lymphatic is marked by double arrows. Arrows point to intact valve trunk afferent to the damaged collector. **e** Phytoparasitic nematode within the lymphatic collector has weakly stained gastrointestinal tract that begins with the stylet, negatively stained stylet knobs, and the esophageal *metacorpus*[63]. Sections thickness: Top: 25 μm, Middle and Bottom: 0.75 μm. Double arrows point to the wall of the collector. In (**c**) and (**e**), Z indicates the relative distance between optical cross-sections. Source data for the graph shown in (**a**) is provided as a source data file. Scale bars: 20 μm, 10 μm in (**c**) insets (YZ cross-sections)

dissemination also preferred by other nematode parasites (Supplementary Discussion).

## Discussion

Here, we showed that the critical filaria adaptation for parasitism, that is, the ability for systemic dissemination via lymphatic circulation is achieved by traits ancestral thus shared by filariae, free-living nematodes, and plant parasites. Therefore, lymphatic vasculature appeared as Achilles' heel of the host, permitting systemic dissemination to any size-compatible nematode parasite.

Most helminths use nonspecific physical or chemical cues such as temperature, oxygen, $CO_2$ tension, and moisture to direct their migration and behavior before entering a host[33,34]. However, the cues that govern host identification cannot explain migratory decisions of a helminth once it enters host tissues. For example, the finding that *Strongyloides stercoralis* detects urocanic acid, the first identified host-specific chemoattractant that helps to locate human skin, does not explain the convoluted routes used by this parasite within a host[35]. In fact, no mechanism or physiological guide used by parasites to complete their migration within host tissues has previously been identified[1,11,36]. On the other hand, the hypothesis linking migration of parasites with the currents of tissue fluids, which was initially proposed by Sukhedo[36], has never been verified[10,12]. The currents hypothesis is attractive owing to its universalism; every tissue in the vertebrate body

generates gradients of fluid currents[37,38], and blood, lymph, and interstitial fluids are the most likely candidates for steering the migration of parasitic larvae that entered the skin. Fluid currents-assisted larval navigation is an example of a fixed action pattern, which could evolve in free-living nematodes as an automated neuromuscular adaptation to existing physical gradients[11]. Therefore, nematodes that respond to currents could be universally adapted to distinct habitats such as, soil or a vertebrate tissues, without the need for new adaptations that match, for instance, species-specific chemokine gradients[19] or sequential proteolysis of specific extracellular matrices[39] (Supplementary Discussion). Indeed, we found that filariae migrating in the microfluidic maze followed the direction of flow at the velocities comparable to the speeds of in-lymphatic migration of filaria larva and the soil-derived nematode. In host tissues, the weak interstitial currents might deter parasites from migrating towards blood vessels[40] rather than directing them toward lymphatic collectors, which do not permeate inward flow of tissue fluid[41,42]. However, the positive current tropism might be more pronounced within lymphatics, where the stronger lymph flow could assist a larva in determining the direction of lymphatic drainage during the lymphatic invasion, and later support the efferent migration toward the lymph node, and further, to blood circulation (Supplementary Discussion).

In sleeping, restrained or anesthetized animals lymph currents drop to minimal levels[29–31], which allows distinction of the effect of lymphatics morphology from the function they control. The valve assistance in the unidirectional, toward the lymph node migration, was highlighted by the entrapment of filaria preferentially at the efferent (sinus) side of valves. The asymmetric accumulation of larvae is likely favored by the asymmetric transverse profile of the valve[42,43], which assures a higher probability for valve crossing when a parasite is probing for the lumen from its narrow trunk side.

Mechanical properties of the dermis together with the anatomy of its vessels might also explain the preference of nematodes for invading lymphatic precollectors as oppose to blood vessels (Supplementary Discussion). Firstly, in contrast to dermal blood capillaries (diameter 6 μm)[44], the large lymphatic precollectors (diameter 43 (SD 15 μm) in mouse skin, 30–100 μm in humans[44]) can enclose 10–25 μm larvae but also provide a space for their sinusoidal bending movement. At the same time, precollecting lymphatics lack the support of adventitia and elastic tunica media, the structural layers found in comparable in diameter veins and arteries[40,44]. Therefore, as the compliance (or stiffness) of thin-walled precollectors is comparable to the stiffness (or compliance) of surrounding interstitium, the lymphatic wall might yield to the invading parasite that attacks the vessel from within the interstitium with a force that depends on the supports, that is, the compliance of that interstitium. According to that mechanical principium[45], without morphological adaptations, a nematode could not penetrate the wall of comparable in size but stiffer arterioles or venules, moving in from the more compliant environment of the interstitium.

Despite the novelty of our observations, the conclusions obtained using infective larvae of *L. sigmodontis* were merely a correlative indication of the mechanism of infection specific to a particular filaria species. If our hypothesis of stochastic wander and lymphatic break-in mechanism was correct, even nonparasitic terrestrial nematodes that are naturally deprived of any vertebrate-specific navigation skills should be able to migrate in the skin and enter lymphatic vessels as long as they can fit and move within the targeted vessel (Supplementary Note 13). Indeed, soil-derived nematodes could migrate in the dermis with multimodal velocities that matched the velocities of filaria migration in the skin and the lymphatics, while nematodes of comparable to

filaria diameter were able to enter lymphatic precollectors. Also, similar to filariae, free-living and phytoparasitic nematodes inflicted various degrees of injuries to collecting lymphatics during vessel invasion, from a puncture wound to the damage that obliterated the continuity of the precollecting vessels.

Since soil-derived nematodes behaved in the skin in a manner similar to specialized obligatory filaria parasites, it is feasible to speculate that other nematodes with an obligatory or facultative parasitic phase are also intrinsically capable of disseminating throughout a body via the lymphatic system. Considering, lymphatics are the probable initial migratory route of the transient human parasite *S. stercoralis*, a free-living nematode with a facultative parasitic phase that begins when larvae enter the skin and end its migratory route at the first microcirculation of the lungs[8]. However, particularly interesting is the confound migratory route of the small (adult diameter 15–29 μm) free-living *Halicephalobus gingivalis*[46]. This soil-dwelling free-living bacterivore causes opportunistic equine meningoencephalitis and nephritis once its parthenogenetic females enter gum or skin lesions. In rarely diagnosed human cases, *H. gingivalis* penetrates the damaged skin and migrates to the brain meninges causing their fatal infection[47]. As the diagnosis of *H. gingivalis* can only be made postmortem, its migratory route remains hypothesized. Even so, lymphatics are a probable portal of systemic entry as necropsy revealed its presence in lymph nodes[48].

Our results suggest that initial dissemination route of skin-invading filariae, but possibly other free-living[7] or zoonotic[8] parasites can be attributed to the set of universal traits that allow nematodes from different habitats to move through the exotic environments of the host tissues, and preferentially invade lymphatics. Such an inherent potential that guarantees a systemic spread of a parasite in unrelated host species would streamline the acquisition of apomorphic and host-specific adaptations, such as antigen mimicry, or anti-inflammatory or tolerogenic responses[49] (Supplementary Discussion). Furthermore, the fact that seemingly harmless but ubiquitous nematodes can in favorable circumstances asymptomatically migrate through the dermis and injure collecting lymphatics suggest the possibility that such overlooked opportunistic nematode infections might contribute to idiopathic or controversial etiologies of various skin-related diseases, such as, podoconiosis (nonfilarial elephantiasis)[4], panniculitides (fat inflammations)[50], or the secondary lymphedema[42].

## Methods

**Ethics statement**. This study complied with all relevant ethical regulations for animal testing and research was carried out in accordance with the Swiss Animal Protection Act and approved by Commission de Surveillance de l'Etat de Vaudois and the Veterinary Authority of the Institutional Animal Care and Use Committee of the University of Chicago. Filariae culture was carried out in accordance with the EU Directive 2010/63/UE and the French "Décret no. 2013-118, 1er février 2013, Ministère de l'Agriculture, de l'Agroalimentaire et de la Forêt.

**Mice**. C57BL/6, C3H, and Chy3/C3H heterozygous mice were purchased from the Jackson Laboratory and bred in the specific-pathogen-free facility. For experiments with the Chy3 mice, 3–9-month-old mice with phenotypic hind paw edema and their aphenotypic littermates were used. The C57BL/6 mice were used at 10–14 weeks old for the secondary tail lymphedema model. Female BALB/c (Charles River Labs) were used at 12–20 weeks old for all intravital imaging experiments with filariae and nematodes. A comparison between lymphatic endothelial cell reporter protein and lymphatic BM staining was made with a Prox1-Cre-tdTomato mouse[51] (Supplementary Methods).

**Maintenance of L3 infective filariae**. *L. sigmodontis* filaria was used as a mixed mouse model of lymphatic and serous cavity filariasis[15,16,52]. Filariae were maintained as previously described[27,53]. Briefly, the patent filaria infection was established in the gerbil *Meriones unguiculatus* in the Muséum national d'Histoire naturelle laboratory in Paris, France. Infective third-stage (L3) larvae of *L. sigmodontis* were dissected from the mite vector *Ornithonyssus bacoti*, counted, and

placed in cell culture medium (H-DMEM; 4.5 g glucose/L, DMEM with 25 mM HEPES, 400 U/ml penicillin, 400 U/ml streptomycin, 1 μg/ml amphotericin B, pH 7.5), and cultured in 5% $CO_2$ at 37 °C. Before staining, L3 larvae were washed four times in cell culture media (DMEM/25 mM HEPES), twice in phosphate-buffered saline (PBS) and transferred to the reaction vial in 100 μl of PBS.

**Subcutaneous inoculation of filariae.** For both models of lymphedema and whole-body imaging, filariae were injected subcutaneously. Before the injection, larvae were manually counted under a bench stereomicroscope while they were collected with a pipette and then breech-loaded into a U-100 insulin syringe. Despite prior silanization of syringes and needles, approximately 10–20% of larvae stuck to the syringe surface; therefore we manually counted these remaining larvae for the exact determination of inoculated larvae.

Mice with primary thigh lymphedema. 3–7-month old Chy3 mice with palpable lymphedema or their aphenotypic littermates were injected subcutaneously in the shank region of the rear paw with 40 infective L3 larvae in 100 μl of RPMI 1640.

Mice with secondary tail lymphedema. One month after the tail surgery, control mice, sham-operated, and mice with tail lymphedema (tail ø at least double the size of sham-operated mice) were injected subcutaneously with 40 infective L3 larvae in 50 μl of RPMI 1640 (Eurobio, France) distal to the wound on the tail (two-thirds of the tail length measured from its base).

Whole-body imaging. Fifty VivoTag XL 680-stained filaria larvae in 200 μl of Hartmann's solution were injected subcutaneously at the dorsolateral side of the lumbar region.

**Primary lymphedema model.** Chy3 mice[25] was initially phenotyped for the presence of chylous ascites in the peritoneal cavity and edematous hind paws 5–10 days after birth. At 8–12 weeks, the persistence of hind paw edema was verified and we only used those in which the diameter of the leg was at least twice that of their littermates for four consecutive weeks. For the group design see the Replications in Supplementary methods.

**Secondary lymphedema model.** Tail lymphedema was surgically induced using by lymphatic ligation at the tail base[54,55]. One day before the surgery until 1 week after, the drinking water was supplemented with acetaminophen (2.5 mg/ml). Buprenorphine (0.03 mg/ml in 0.9% NaCl) was injected intraperitoneally 1 h before the surgery, and the tail was depilated with Veet® cream. Two circumferential incisions, half a centimeter apart, were made through the dermis of the mouse tail at approximately one-third of the tail length, measured from the tail base. The resulting skin ring was carefully cut away from the subcutaneous tendons and major blood vessels (or, in control sham-operated mice, the skin is not removed). Then, a 1-cm-long silicon tube was placed over the exposed wound, and the wound area was covered with Bethaphen cream and dressed with multiple layers of a sterile bandage. The dressing was fastened in place with a tight-fitting, 4-cm-long plastic tube cut from a 50-ml Falcon® serological pipette with a hot knife (Supplementary Fig. 6a). The tube was prevented from sliding off the tail during regular mouse activity by gluing multiple layers of tape to the tip of the tail. The mounted tube could freely slide towards the base of the tail, which allowed wound inspection and dressing change in days following the surgery. Mice with the secured dressing were returned to their cages (one mouse per cage), and the dressing was changed daily until significant bleeding was no longer observed (usually 2–5 days post-surgery; Supplementary Fig. 6b). The mice were then monitored twice a day, and the dressing was changed every 3 days. Mice with sustained tail swelling, with the diameter at least twice that of sham-operated mice at 30 days, were used for the experiments. For the group design see the Replications in Supplementary Methods.

**Infection efficiency of *L. sigmodontis* larvae in lymphedema.** Ten days after injection, mice were euthanized, and the pleural cavity was lavaged with ten washes of 1 ml PBS. Larvae recovered from the pleural cavity were counted under the phase-contrast microscope, and the infection efficiency was expressed as a percentage of inoculated to recovered filariae.

**Intravital imaging for whole-body distribution of filariae.** The in vivo distribution of inoculated larvae was imaged using the 2D capabilities of a Visen Fluorescence Molecular Tomography 2500 device (FMT 2500, Flagship Pioneering). Larvae were manually counted under bench stereomicroscope while they were collected with a pipette and breech-loaded into a U-100 insulin syringe. 50 filaria larvae stained with near-infrared (VivoTag XL 680, PerkinElmer, Waltham, MA) in 200 μl of Hartman's solution were injected s.c. at the dorsolateral side of the lumbar region. Mice were then either imaged continuously every 3–5 min for 2 h, or imaged once and returned to the cage for 5 h, when it was anesthetized and imaged again. Mice were imaged again at 18 h and then euthanized. To account for the systemic escape of larvae, individual filariae were counted at different time points. Larvae that could not be accounted within the imaging area of the skin on the following time point were considered as ones that escaped the skin into the systemic circulation. Data were collected using TrueQuant software (PerkinElmer), and images were exported as high-quality jpeg files. This experiment was done twice with 50 filaria larvae inoculated each time.

**Isolation of soil-derived nematodes.** Free-living and plant parasitic nematodes, were isolated from soil samples collected at the University of Chicago, the monitored location not visited by the wild, household, or farm animals. Actively moving nematodes were purified from the soil samples using the Baermann funnel method[56]. Briefly, large particles were first removed from 100-g soil samples by mixing soil-samples with 300 ml of deionized water and passing the slurry through a funnel-shaped No. 20 sieve (850 μm mesh size). The same mesh sieve served as a screen after it was placed within a larger funnel fitted with 5 cm of rubber tubing, which was sealed with a hemostat. A Kleenex tissue paper was placed on the screen, and deionized water was poured into the funnel to cover approximately half of the fitted screen. Soil samples were then poured onto the screen, and nematodes were collected in the first 50-ml fraction from the rubber tubing 48 h later. Nematodes were spun down at 500 g and hand-picked with a 200 μl pipette from the residual 5 ml of water. Collected nematodes were kept in deionized water at 4 °C for up to 24 h. Prior to the staining with tetramethylrhodamine isothiocyanate (TRITC), the nematodes were washed twice with PBS, pH 7.4, and transferred to the reaction vial with 100 μl PBS.

**Imaging of filariae and soil-derived nematodes.** Due to lack of prior expectations that would warrant a test of the preliminary hypothesis of filariae and soil-derived nematodes behavior in the skin, the imaging of their migration was performed as an observational study, therefore the size of experimental groups could not be predetermined[57].

**Intradermal (dorsal ear) inoculation of filariae and nematodes.** Nematodes were manually counted using a bench stereomicroscope with a pipette and benchtop handheld counter. For the intradermal inoculation[58], 40–50 TRITC-stained nematodes were transferred to Hartmann's solution and muzzle-loaded into the needle-holding chamber of the 50 μl Hamilton syringe. During the injection (3–5 min) nematodes were inoculated in 15–25 μl of Hartman's solution while retracting the needle. This volume was sufficient to inoculate approximately 30–40 nematodes in the postneedle space (Supplementary Fig. 7). Only one ear was used for inoculation of filariae or nematodes.

**Labeling of L3 larvae and soil-derived nematodes.** Unless stated otherwise procedures described here were identical for soil-derived nematodes and filaria L3 larvae (collectively referred to as nematodes). Prior to the labeling, 25 mg/ml TRITC stock in dimethyl sulfoxide was diluted 20 × in PBS, and after vortexing, the solution was passed through a 0.45-μm filter. The 100 μl of the soluble fraction of TRITC in PBS was immediately mixed with 100 μl of PBS containing 100–300 nematodes for 12 min. Alternatively, VivoTag 680 XL NHS ester diluted in PBS at a final concentration of 1 mg/ml was used to label L3 larvae for whole-body near-infrared imaging in a Visen imager (Flagship Pioneering). As soon as nematodes were mixed with fluorescent amine-reactive reagents, they were kept in the dark. The reaction was stopped 12 min later with 1 ml of bovine serum. Labeled nematodes were washed five more times; filaria larvae were then washed twice with 10% fetal calf serum (FCS) in DMEM/HEPES and kept at 37 °C in cell culture incubator while soil-derived nematodes were additionally washed twice with 10% FCS in deionized water and kept at 4 °C.

**Intravital imaging of intradermally inoculated filariae.** Intravital skin staining in the ear[20,21,59] was performed to label lymphatic vessels before inoculating nematodes. The lymphatic reporter mouse (Prox1-Cre-tdTomato[51]) was not used because of the incomplete penetrance of the Prox1-tdTomato reporter protein in fragments of lymphatic vasculature (Supplementary Fig. 8) as well as the signal overlap with the TRITC-labeled nematodes (Supplementary methods). For long-term (up to 12 h) intravital imaging, mice were initially anesthetized with 50 mg/kg ketamine and 10 mg/kg xylazine, and after 2 h injectable anesthesia was gradually replaced with an inhalable isoflurane/oxygen mix (from 0.5 to 1.5%; Minrad Inc., Buffalo, NY) that was humidified through a Drechsler's bottle. Mice remained anesthetized until they were euthanized by $CO_2$ asphyxiation and intracardiac fix-perfusion.

Ventral skin and cartilage at the proximal and central location of the ear were detached from the dorsal skin flap, leaving approximately 20% of the distal ear intact. The exposed proximal dorsal skin was then stained for 10 min with 10 μg/ml of biotinylated anti-collagen IV (Abcam, ab6586) or anti-Lyve1 (ReliaTech, 103-PA50) antibodies in Hartman's solution supplemented with 2 mg/ml aprotinin. The labelling was detected with streptavidin-Alexa 488 (Invitrogen, 711-546-152) or anti-rabbit-Alexa 488 F(ab')2 (Jackson ImmunoResearch Laboratories, 711-546-152) antibodies. After five additional rinses in Hartman's solution, 35–50 larvae were then injected intradermally in the tip of the ear in the dorsal dermis; the location protected by cartilage and ventral skin fragment (distally to the exposed and collagen IV-stained dorsal skin fragment). Immediately after, the ear was mounted under a fluorescence stereomicroscope (M250 FA, Leica Microsystems CMS GmbH, Wetzlar, Germany), immersed in 150 mM freshly prepared sodium ascorbate and 15 mM $CaCl_2$ and imaged in the Texas Red (filariae) and Red2 (collagen IV) channels. After imaging (2–12 h), the mouse was perfused with Zinc fixative.

**Intravital imaging of intradermally inoculated soil-derived nematodes**. Cutaneous migration of soil-derived nematodes was imaged through the intact skin, without surgical preparation and intravital immunostaining. Totally, 30–50 TRITC-labeled nematodes were injected in the dorsal skin at the tip of the ear and imaged as above. The imaging was paused when active nematodes stopped moving, which we considered a sign of extensive phototoxicity. After imaging, mice were perfused with zinc fixative and the dorsal skin flap was stained for collagen IV. A single, long-distance migratory path of a nematode was later matched to the lymphatic precollector network reconstructed from individual maximum intensity projection images of the collagen IV staining.

**Whole-mount preparations and confocal imaging**. In some mice, a spread of TRITC-labeled filariae or soil-derived nematodes within dermis were imaged from 18 min to 3 h after inoculation in the dorsal ear dermis. Mice were suffocated in $CO_2$, and systemic blood was removed through cardiac perfusion (95–100 mm Hg) with 25 ml of Hartman's solution supplemented with 25 mM HEPES and 0.1% glucose. After complete exsanguination of the animal, Hartman's solution was replaced with 25 ml of Zn fixer (39 mM $Zn(CH_3COO)_2$, 37 mM $ZnCl_2$, 4.5 mM $CaCl_2$, 50 mM Tris, pH 6). Ears and draining superficial cervical LNs[60] were immediately dissected and placed in ice-cold Zinc fixer with 1% Triton-X-100 and 5% DMSO for 24 h at 4 °C. After that, dorsal and ventral skin flaps were separated, and skin flaps were returned to fixative for another 24 h. Since soil-derived nematodes are resistant to zinc fixative, ear skin flaps on those mice were separated immediately after intracardiac perfusion. Then, after two washes in ice-cold TBS (25 mM Tris, 140 mM NaCl, pH 7.5), nematodes in the dorsal skin were killed with a 10-min incubation of the dorsal skin in 1% $NaN_3$ in TBS. After two washes in TBS, skin flaps were returned to Zinc fixative (with 1% Triton-X-100 and 5% DMSO) for another 24 h. Tissues were then washed twice in TBS with 0.1% Tween, once in TBS, blocked for 15 min in 0.5% casein in TBS, and incubated for 24 h with 10 μg/ml of primary antibody: biotinylated rabbit anti-collagen IV, rat-anti VE-cadherin (B&D, 550548, clone 7B4), hamster anti-CD31 (PECAM-1, Genetex, clone 2H8, GTX74943), or rabbit anti-Lyve1 in 0.5% casein in TBS. After being washed in TBS 0.1% Tween, tissues were incubated with the appropriate secondary antibody from Invitrogen: donkey anti-rabbit Alexa-488 (A21206), goat anti-hamster Alexa-488 (A21110) or donkey anti-rat Alexa-594 (A21209), and streptavidin-Alexa 647 (Invitrogen, A21110) in 0.5% casein in TBS for 24 h. Washed tissues were immersed in Murray clear (1:2 benzyl alcohol/benzyl benzoate (Fisher, 99% pure)) with 25 mM propyl gallate, mounted on a glass slide, and imaged in a Leica SP5 confocal microscope equipped with a white light laser or Olympus IX82 microscope equipped with Olympus disk spinning unit, Spectra X light engine from Lumencor and Orca-Flash 4.0 v2 from Hamamatsu. Optical sections, maximum intensity projections (MIPs) or overlay surface masks were generated from image stacks using Imaris 7.1 (Bitplane AG, Zürich, Switzerland). See Supplementary methods for technical details of imaging and image processing.

**In vitro migration of filariae in a microfluidic maze**. To determine the effect of fluid currents on the directionality and speed of filariae migration, we developed a microfluidic maze that allowed generation of a broad range of defined fluid currents (Fig. 4a). The chip was composed of a network of $50 \times 50$ μm square microchannels and three fluidic ports: two (at NE and SW corners) to drive the perfusion, and a Central Station (C-S) for introducing larvae. C-S was sealed before initiating flow (Fig. 4b). The angles between the channels were all 60° so that the direction of larval migration would have no geometrical bias.

The microfluidic chip was developed using a soft lithography technique[61]. Briefly, the architecture was designed with CleWin (WieWeb Software, Hengelo, NL) and printed on a chrome mask using a high-definition laser (DWL200, Heidelberg). The negative photoresist 1050 was spin-coated on 100-mm silicon wafers and patterned using a mask aligner (MA6/BA6, Süss). A PDMS prepolymer and curing agent (Sylgard 184, Dow Corning) mixture was mixed according to the manufacturer's instructions, degassed, poured on the silicon master, and cured at 80 °C for 1 h. The PDMS cast was peeled off the wafer, and the fluidic ports were punched. The device was sealed with a glass coverslip after oxygen plasma treatment and filled with cell culture media (25 mM HEPES, bicarbonate-free RPMI 1640 with 10% FBS, Gibco).

Larvae were pipetted into the C-S in bicarbonate-free DMEM buffered with 25 mM HEPES and allowed to spread throughout the network for 10 min before starting flow or imaging. For the flow condition, 15 nL/s was pumped into the northeast corner, with the opposite southwest port serving as the outlet. Larval migration was recorded for 30 min, and their movements were then manually tracked (Fig. 4b). The positions and trajectories were then correlated with the theoretical velocity fields of flowing medium, computed with commercial CFD software (Comsol Inc.; Fig. 4b).

To express the directionality of larvae migration, we calculated the relative net displacement vector $R_f$ as the dot product of the net displacement vector $R$ (connecting the starting and end points of the larval trajectory) with the net flow velocity vector (Fig. 4c).The following experiments were performed: First experimental day: static (7 larvae), flow (2 larvae), static (2 larvae), flow (23 larvae), flow (7 larvae). On the other occasion, the static experiments were performed once with 6 larvae, and the flow experiment twice with 5 and approximately 45 larvae.

**In vitro pocket entrapment of filariae**. Unlabeled filaria larvae were kept in 10% FCS in DMEM with 25 mM HEPES, pH 7.5. 10 larvae were taken to a separate vial, and 1 mg/ml of lignocaine was added for 10 min. After that, 10 paralyzed larvae and 10 freely moving larvae were briefly washed in Hartman's solution and mixed in 3 mg/ml rat tail collagen type I neutralized to pH 7.5 with 100 mM NaOH, 10× concentrated bicarbonate-free DMEM and 25 mM HEPES, pH 7.5 at 4 °C (2.4 mg/ml final concentration of collagen). Collagen sol was allowed to polymerize for 10 min at 37 °C and was covered with 1.5 ml of 10% FCS in HEPES-DMEM media. The location that overlapped a paralyzed larva and moving larvae were selected for continuous bright-field imaging in Leica stereomicroscope.

**Statistical analysis**. Except for Fig. 1a, tests were computed with the assumption of two-tailed distribution. The nonparametric Mann–Whitney $U$ test was used to compare datasets with $n < 10$. Multiple comparisons of three nonparasitic groups of nematodes with L3 filariae were done with Kruskal–Wallis rank test with Dunn's correction. Comparison of larger groups was made with the non-paired $t$ test when datasets passed the normality test. Mean velocities were compared using an unpaired t-test or one-sample t-test by comparing means to the maximum speed of filariae or nematode migration within lymphatics calculated in Fig. 2d or Fig. 5a. The unidirectionality of larvae migration within the microfluidic maze, separately for static and flow conditions, were tested with one-sample $t$ test with null hypothesis assuming random migration of filariae ($\overline{Rf} = 0$) using GraphPad "QuickCalcs-OneSampleT1", https://www.graphpad.com/quickcalcs/OneSampleT1.cfm (accessed 5 Oct 2018). The effect of the excision of lymphatic collectors in the secondary lymphedema model on the filariae migration to the pleural cavity was tested with Fisher exact test (binominal values from two pooled experiments) using $2 \times 3$ table calculator from SISA (Daan G Uitenbroek, "SISA-Fiveby-Two," 1997. https://www.quantitativeskills.com/sisa/distributions/binomial.htm (accessed 21 Feb 2019). The asymmetrical co-localization of filariae and nematodes at the valve inlets (afferent trunk) or outlets (efferent sinus) was tested with the sign test (with H0 being the assumption that the distribution at trunk and sinus localizations had equal (0.5) probabilities; GraphPad QuickCalcs-Binominal, https://www.graphpad.com/quickcalcs/binomial1/ (accessed 21 Feb 2019)). Means in the text and graphs are presented, respectively, as SD () or bars of corresponding standard deviations. The effect size was calculated for the parametric comparisons. For the effect size, the standard deviation of pooled population was estimated from pooled standard deviations of the compared groups using Cohen's index, (2) $SD_{pooled} = \sqrt{\frac{(SD_1^2 + SD_2^2)}{2}}$. With two exceptions noted above, statistical calculations were done using Prism 8.0.1 (GraphPad, La Jolla, CA).

**Reporting summary**. Further information on research design is available in the Nature Research Reporting Summary linked to this article.

## Data availability
All data generated or analyzed during this study are included in this published article (and its Supplementary information files). The source data underlying Figs. 1a, b, 2b, d, 4d, and 5a are provided as a Source Data file. Raw data are available from the corresponding author upon reasonable request.

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

## Acknowledgements

The authors are grateful to J. Kilarska and E. Güç for help with image processing and to V. Borel and M. Pasquier for technical assistance with secondary lymphedema model. This project was supported by grants from the European Research Commission (206653-2 to M.A.S.).

## Author contributions

W.W.K. designed and performed the experiments, analyzed the data, and wrote the paper; M.P., C.M. and O.B. performed and designed the experiments; S.B. designed the experiments and M.A.S. designed the experiments and wrote the paper.

## Additional information

**Competing interests:** W.W.K. has a patent pending (Application no. 62/769,579) for the nematode labeling methods used in this report. Remaining authors declare no competing interests.

