## [Peer Review File · Nature Communications]

Editorial Note: Parts of this Peer Review File have been redacted as indicated to avoid misconception.

Reviewers' Comments:

Reviewer #1:

Remarks to the Author:

General Comments:

In this study, the authors label L3 infectious stage filarial larvae with fluorescent dyes and then track their movement by evaluating them with a range of imaging techniques including whole body infrared imaging, confocal microscopy, and intravital microscopy. Additionally they performed studies in animals with dysfunctional lymphatic collectors and in vitro studies using microfluidic mazes.

This study is a breakthrough article with regards to our understanding of some of the first steps in mammalian host invasion by filariae. The first experiment alone makes a finding that is substantial: *Litomosoides sigmodontis* appears to predominantly use lymphatic vessels to migrate to the pleural cavity. While there are numerous histological studies showing L3s rapidly "escape" into lymphatic vessels after entering the dermis and subcutaneous tissues, this is the first study to truly demonstrate that this is a required route for L3 larvae.

One of the most impressive videos is that of Video 4, which shows an L3 forcefully puncturing and entering a lymphatic vessel. The accompanying figure (Fig 3d) shows clearly how the worm was unable to penetrate a blood vessel capillary but was able to penetrate the more thin-walled lymphatic capillary. The tissue destruction seen in a lymphatic vessel where an L3 entered (figure 3B) is also impressive and also demonstrates a forceful, destructive entry by L3s into lymphatic vessels.

Other important findings include: observation of slow migration in tissues followed by much more rapid transit in lymphatics, the presence of biphasic movement in both tissues and lymphatic vessels (inactivity periods followed by bursts of movement), and ability of L3s to reach the subcapsular sinus of a draining lymph node less than 20 minutes after injection. Finally, the authors also show results with regards to the unidirectional movement of the larvae. The authors show images that are consistent with the notion that valvular anatomy helps direct filarial toward the lymph nodes. They later utilize microfluidic mazes to show that larvae move along the direction of fluid flow.

Major critiques: none

Minor critiques

1. In Supplementary figure 1...it would be helpful to have a graphic of a mouse highlighting the area being photographed to orient the reader.
2. In supplementary figure 1...we still see a lot of signal at 20 hours. Do these represent L3 larvae that became trapped in the dermis and are now dying and will never successfully migrate? Might be nice to comment on what this late signal may mean in the results or discussion.
3. Supplementary figure 2 is remarkable, but I'm not sure exactly which vessels the L3 followed. Is the yellow line to the right of the vessels through which the L3 migrated, or right on top? It might be easier to simply put two (composite) images side by side, one with the yellow path and one without. Also, on figure 4 it would be nice to see a graphic of the ear to give the reader a sense from what point to what point on the ear the L3 was imaged.

Reviewer #2:

Remarks to the Author:

The manuscript by Kilarski et al documents imaging experiments that were performed in mice to visualize the entry and movement of filarial larvae in the lymphatic system. The authors use a novel imaging approach involving fluorescent labeling of worms and in vivo whole mount staining of mouse tissue to record videos of the infiltration of the parasite and conclude that tissue forces are playing a role in the larvae spread. While the imaging results in some very impressive images and videos, the data are presented in an unclear and anecdotal fashion and it is difficult to appreciate what the

potential impact of such a study could be. In many cases, important experimental details are missing and interpretations are made without enough supporting evidence.

Major points:

1. What is the relevance of such a study? While the acquisition of images of the movement of the worms into and through the lymphatic vessels is interesting, how would this information be used to stop such an invasion in human beings? The authors in the discussion touch upon previous studies where the "cues that govern host identification and entry of various helminths is invaluable from prevention and epidemiological perspectives." It is not clear how the knowledge that was gained from the current study would contribute in a similar manner.
2. Lack of experimental details – this was an issue throughout the manuscript. The most egregious example of this are the very first figures (Figure 1A and 1B). There are absolutely no details in the method section for these experiments. How were the larvae counted in the pleural cavity? At what time point after injection were they counted? How were the surgeries performed? How soon after surgery were the larvae injected? Why were there only 4 mice in the surgery induced lymphedema group, while there were 8 in the sham surgery group? Were any statistical analyses performed?
3. Many claims are made from anecdotal or insufficient evidence – the velocity measurements that were generated for larvae movement through the interstitial space and within the lymphatic vessels appear to have been made from only one video in each situation. The directional migration of filarial larvae was inferred from the position of the head and tail from static images, however, dynamic statements of filarial motion were made at this point – "filarial tail to push off against the vessel wall, accelerating larval downstream movement". An in depth analysis of the lymphatic wall breach was performed based on n=1 experiments, as well as the interpretation of the activity of the one larva that was captured entering a collecting lymphatic vessel.

Minor points:

1. Why were the experiments not performed in lymphatic reporter line, such as Prox1-GFP mice? This would allow better visualization of the lymphatics without having to resort to interpretation of the morphology from the in vivo whole mount approach using collagen IV.
2. Why do the authors make several statements explaining why ear skin is not the best tissue for inoculation of larvae (pg. 6 line 133 to pg. 7 line 144) and then proceed to use ear skin for all the remaining in vivo experiments?
3. The authors state that lymph node channels impede further movement of the larvae, but if this were true, how did the larvae reach the pleural cavity in the initial experiments?
4. Figure 4C and 4D do not appear to be discussed anywhere in the results.
5. The interstitial flow mechanism is greatly weakened (as acknowledged by the authors themselves in the discussion) by the conclusion that the larvae entered at the lymphatic collectors, rather than the capillaries.
6. The discussion mentions that larvae could move forcefully against the lymph current – where was this shown?
7. The discussion regarding chemokine gradients and lateral gene transfer is extraneous and does not appear to be related to the presented data.
8. The text contains many spelling errors (e.g. "lymphodema", collagen "sol", "colecting", "as a mouse on egg chew") and other mistakes throughout

Reviewer #3:

Remarks to the Author:

Mechanical cues guide lymphatic migration of filaria *Litomosoides sigmodontis* during initial phase of infection by Kilarski et.al.,

One of the questions that has been largely been unanswered in the filarial infections is the mechanism(s) of tissue homing of the infective larvae. While several chemotactic factors, nutritional needs etc. have been proposed, they could not be tested efficiently. In this regard using *Litomosoides* model, provides a good advance in our understanding some of the early events and generate new hypotheses. The use of brute force entry into the collecting vessels is really nice.

1. It is understandable that sometimes it is better to add in pieces of discussion in the results to help the flow and tell a better story. However, it appears that there is a lot more discussion within the results that can be moved or restricted.
2. Except for 14 inoculated ears, there is no mention of the number of mice and the background used in the methodology (except for the Chy-3 with C57BL/6).
3. The demonstration of the collecting vessels is nice. However, as noted by the authors different filariae home to different tissues. Because this model is limited to the very initial phases and not likely dependent on host susceptibility of other filarial parasites, it would have been more informative to have use other filarial larvae that are easily available such as *Brugia malayi* and *Dirofilaria immitis* and have different tissue specificities.
4. The understanding in the field is that the L3's are not 'injected' into the host by the vector. Rather they get released onto the skin near the bite wound and find their way into the host through the bite wound. So it is not clear or how helpful the entrapment analogy can be used to explain the egress of the larvae. I wish there was a way to stain the larvae 'in situ' in mites, that can then be used for infecting mice.
5. Not all L3's that enter the host develop into adults. The authors show the varying number of L3s in the collecting vessel after injecting ~30 L3s. Is there a quantitative data of the number of L3's successfully in the collecting vessels compared to L3's stuck in tissue(s)?
6. Is the inability of the L3's to penetrate the smaller blood vessels related to the vessel architecture? Not sure if it was discussed.

Minor comments

1. Pg 6 Ln 132: The labeling for 1D-H was stained for collagen. Should it be stained for VE cadherin and collagen?
2. Supplementary Figure 3 to me was not clear. It would be helpful if there was a cartoon to help understand the anatomical locations being represented.
3. Typographical errors need to be addressed
7. Pg 6 Lns 118-122 – Is this related to finding the orientation of the worms?

Dear Editors and Reviewers,

We thank you and the Reviewers for the critical and helpful evaluations of our manuscript: "*Mechanical cues guide lymphatic migration of filaria Litomosoides sigmodontis during the initial phase of infection.*" In accordance with Reviewers' critique, large parts of the manuscripts were re-written to place a more direct focus on our results, while the additional discussions were transferred to the 'Supplementary methods and discussion' file. We corrected various errors and omissions, also mistakes that were not noted by the Reviewers. For example, due to the specific handling of metadata by the Leica LSM software, we had mistakenly underestimated the speed of filaria migration in the lymphatics. Additionally, we present new data (new Video 1, 3, 4, 6, 9) that, for instance, allowed us to calculate the average velocities of filaria migrating in the tissue. Also, we added the average velocities of filaria migrating in the microfluidic maze under static and flow conditions. With the descriptive statistics, when possible, we computed the inferential statistics. Finally, the new paragraph 'Terms used' in the Method section was intended to clarify the various terms used in the manuscript.

We also apologize for the missing parts of the 'Materials and methods' section. The corrected Methods include new information that was missing in the original manuscript: Secondary lymphedema model and Statistics. Additionally, we added a Technical Figure 1, which depicts the most critical changes we made to the secondary lymphedema model as compared to the published methodology and Technical Figure 2, in which we compare the basement membrane staining with the lymphatic-restricted expression of the tdTomato fluorescent protein. Supplementary Figures 1 to 3, have now diagrams explaining the experimental procedures. Finally, also in response to Reviewers' questions, we discuss the technicalities of each experiment in the Result section.

Most importantly, we have added an entirely new set of imaging experiments with intradermally injected TRITC-labeled free-living nematodes. These experiments (new Fig. 5, new Supplementary Fig. 5, and six new videos: Video 11, 12, 13A, 13B, 14A, 14B, and 15) show that even without evolutionary adaptations, free-living (non-(zoo)parasitic) nematodes are able to migrate within the skin and then localize and enter lymphatics. We found that similar to filaria parasites, free-living bacterivorous and plant-parasitic nematodes were able to forcefully enter lymphatic collectors. Due to specifics of the soil-isolated free-living nematodes (i.e., substantial variation in size and movement pattern), we were unable to repeat the migration experiments precisely as we did with filaria larvae, that is, in the pre-labeled skin. Instead, we imaged nematode movement in the dermis through the intact skin. Nevertheless, we show that the skin migration of the free-living nematodes, similar to the migration of parasitic filariae, was multimodal, with two distinct speeds that matched the magnitudes of filarial velocities in the dermis and the lymphatics. Similar to filariae, free-living nematodes could enter collecting lymphatics by inflicting the various level of damage to the vessel. Therefore, likely without the necessity for chemotactic guidance or proteolysis of the extracellular matrix, free-living nematodes were able to migrate in the skin and enter lymphatics in a manner similar to specialized obligatory filarial parasites. This observation proved our original hypothesis that no host-specific adaptations are required for the dissipation of parasitic nematodes throughout the tissues of a vertebrate organism.

Finally, as the new results generalized the conclusions of our manuscript, we decided to modify its title. The new title is "*Inherent properties of nematodes facilitate spread of filaria parasites by exploiting morphological vulnerabilities of lymphatic collecting vessels.*"

The major changes in the modified manuscript are highlighted in yellow (for details, please see the point-by-point response to Reviewers below). We hope that by adding new data and clarifying many

problematic aspects of the original manuscript, you and Reviewers find our responses satisfactory and the manuscript acceptable for publication in *Nature Communications*.

Reviewer #1 (Remarks to the Author):

General Comments:

In this study, the authors label L3 infectious stage filarial larvae with fluorescent dyes and then track their movement by evaluating them with a range of imaging techniques including whole body infrared imaging, confocal microscopy, and intravital microscopy. Additionally they performed studies in animals with dysfunctional lymphatic collectors and in vitro studies using microfluidic mazes.

This study is a breakthrough article with regards to our understanding of some of the first steps in mammalian host invasion by filariae. The first experiment alone makes a finding that is substantial: *Litomosoides sigmodontis* appears to predominantly use lymphatic vessels to migrate to the pleural cavity. While there are numerous histological studies showing L3s rapidly “escape” into lymphatic vessels after entering the dermis and subcutaneous tissues, this is the first study to truly demonstrate that this is a required route for L3 larvae.

One of the most impressive videos is that of Video 4, which shows an L3 forcefully puncturing and entering a lymphatic vessel. The accompanying figure (Fig 3d) shows clearly how the worm was unable to penetrate a blood vessel capillary but was able to penetrate the more thin-walled lymphatic capillary. The tissue destruction seen in a lymphatic vessel where an L3 entered (figure 3B) is also impressive and also demonstrates a forceful, destructive entry by L3s into lymphatic vessels.

Other important findings include: observation of slow migration in tissues followed by much more rapid transit in lymphatics, the presence of biphasic movement in both tissues and lymphatic vessels (inactivity periods followed by bursts of movement), and ability of L3s to reach the subcapsular sinus of a draining lymph node less than 20 minutes after injection. Finally, the authors also show results with regards to the unidirectional movement of the larvae. The authors show images that are consistent with the notion that valvular anatomy helps direct filarial toward the lymph nodes. They later utilize microfluidic mazes to show that larvae move along the direction of fluid flow.

Reviewer #1 (Remarks to the Author):

Minor critiques

1. In Supplementary figure 1...it would be helpful to have a graphic of a mouse highlighting the area being photographed to orient the reader.

A schematic depicting the experimental procedure has been added as panel A to the Supplementary Figure 1. Additionally, schematics has also been added to Supplementary Figures 2 and 3.

2. In supplementary figure 1...we still see a lot of signal at 20 hours. Do these represent L3 larvae that became trapped in the dermis and are now dying and will never successfully migrate? Might be nice to comment on what this late signal may mean in the results or discussion.

The experiment shown in Supplementary Fig. 1 aimed to confirm that fluorescently labeled nematodes are able to dissipate throughout the skin and eventually leave the skin. Supplementary Fig 1 shows that half of the injected larvae could not be accounted for 18 hours after the injection, with over 70% of the infective filariae larvae being able to escape from the injection site within 2 hours and at least 30% of these filariae exited the dermal location within 20 hours post inoculation.

These results are in accordance with the previously published effect of the inoculation by injection on the efficiency of filaria skin escape. Nevertheless, some (up to 10 out of 39 injected minus 29, the maximum number that spread in the dermis) larvae died or were entrapped at the injection site. This was in contrast to the intradermal injection in the ear, where the majority of larvae stay at the injection site. Without Supplementary Fig. 1, one might presume that labeling caused the death or entrapment of larvae at the injection site in the ear (our main experimental tissue location). However, the experiment that is shown in Supplementary Fig. 1 indicated that the intradermal inoculation itself, rather than a chemical modification to filariae, was responsible for their initial entrapment. We discuss the inoculation problem in the new Method section. We also added two videos, new Video 1 and new Video 12, that show entrapment of, otherwise actively moving, nematodes at the intradermal injection site of the ear skin.

3. Supplementary figure 2 is remarkable, but I'm not sure exactly which vessels the L3 followed. Is the yellow line to the right of the vessels through which the L3 migrated, or right on top? It might be easier to simply put two (composite) images side by side, one with the yellow path and one without. Also, on figure 4 it would be nice to see a graphic of the ear to give the reader a sense from what point to what point on the ear the L3 was imaged.

Supplementary Figure 2 was corrected according to the Reviewer's comment. We have added a representative image (Supplementary Fig. 2A) in which we label the path of a lymphatic collector in a similar manner as shown in Supplementary Fig. 2 (now, Supplementary Fig. 2B). The same lymphatic collector, without the yellow line marking the lymphatic path, is shown in Fig. 2E. Also, the complete lymphatic path in Supplementary Fig. 2B is marked with arrowheads.

Reviewer #2 (Remarks to the Author):

The manuscript by Kilarski et al documents imaging experiments that were performed in mice to visualize the entry and movement of filarial larvae in the lymphatic system. The authors use a novel imaging approach involving fluorescent labeling of worms and in vivo whole mount staining of mouse tissue to record videos of the infiltration of the parasite and conclude that tissue forces are playing a role in the larvae spread. While the imaging results in some very impressive images and videos, the data are presented in an unclear and anecdotal fashion and it is difficult to appreciate what the potential impact of such a study could be.

We share the frustration with the Reviewer regarding the scarcity of research that has been done on that vital topic; we hope that our work is only the first step in deciphering the mechanisms of parasite infection. Since the results and ideas presented in our manuscript will be the base for future studies, we anticipate that the methods we described will provide the missing methodological platform for the field to fill the current knowledge gap.

Unfortunately, and we strengthen that notion in our current manuscript, we could only base our research plan and discussion on such 'anecdotal' evidence.

In many cases, important experimental details are missing and interpretations are made without enough supporting evidence.

We apologize the Reviewer for missing experimental details. The Method section has been expanded and is now supported by two Technical Figures and methodological schematics in Supplementary Figures 1, 2, and 3 and Videos 1-4 and Video 11.

Major points:

1. What is the relevance of such a study?

Naturally, only the future will show if the knowledge obtained in this study have any far-reaching consequences with regard to repeated injuries to collecting lymphatics during multiple infections in the endemic area. [redacted]

On the other hand, this research would be invaluable if the goal was to find a new approach to vaccination that aims at the stimulation of cellular (anti-viral, anti-tumoral), instead of humoral immunity. Individually, we can further deliberate on the second option. We are finalizing another manuscript on the use of filaria as a vehicle that could co-deliver a custom antigen with an intrinsic *Wolbachia* adjuvant that can stimulate Th1 responses. Further, because of new results presented in the revised manuscript showing that free-living (saprophytic) nematodes can also enter lymphatic system, we plan to analyze the specific morphological type of these nematodes and use this nematode species as an antigen carrier.

Aside from that, we feel that it is remarkable that damage to the collector during lymphatic entry by the filarial larvae has not been reported previously. The significant damage to the collector remains asymptomatic, but it is possible that it might have long-lasting effects on other pathophysiological processes. Additionally, as we showed in the modified manuscript, the lymphatic path can be used by other roundworms, such as free-living nematodes that cause opportunistic (and likely unreported) infections. Therefore, *“the experimental verification of the initial lymphatic route of parasites such as filariae and H. gingivalis should allow the correlation between morphological parameters of nematodes and their hypothesized habitats. In turn, this might prompt an investigation of their potential role as the etiological factors in various human idiopathies.”* This reasoning has been added to the Discussion section.

We believe that this manuscript might be indeed a significant step forward in fields that are not necessarily linked to the prevention of filariasis. Nevertheless, it is quite difficult to imagine any attempt to stop the filarial invasion in humans without the understanding of the invasion mechanism.

While the acquisition of images of the movement of the worms into and through the lymphatic vessels is interesting, how would this information be used to stop such an invasion in human beings?

As we state in the Introduction, a responsible design of the therapy requires an understanding of the processes that the therapy is aimed to correct or prevent. Therefore, an understanding of the mechanism of filaria invasion is a pre-requisite for treatments that intent to stop the filarial invasion in humans.

Our work aimed to elucidate the mechanisms that govern the varying and often confounding migratory paths of filarial nematodes. That basic knowledge might at one point help to block filarial infection. For example, one could plan the use of two-photon photodynamic therapy to block collecting lymphatics. in front of the lymph node to stop dissipation of parasites. The question remains how fast the treatment must be applied and should lymph node afferent or efferent lymphatics be targeted. Alternatively, should the treatment occlude lymphatics, or whether it would be sufficient to induce intralymphatic hyperplasia (for example with VEGF-C) and block the drainage of large particles, such as filarial larvae, without actually blocking the lymph flow.

These are just a few hypothetical examples where our work provides a starting platform for further research. Nevertheless, we show a practical application of the known local anesthetic, lignocaine that temporarily paralyzes infectious filariae (Video 2). The approach might be applicable to transiently block the spread of parasites from the site of infection.

The authors in the discussion touch upon previous studies where the “cues that govern host identification and entry of various helminths is invaluable from prevention and epidemiological perspectives.”

We apologize to the Reviewer for the unclear statement. The sentence has been rewritten to: *“Although understanding the cues that govern host identification is invaluable from preventive and epidemiological perspectives, this information cannot explain migratory decisions of a helminth once the parasite enters host tissues.”*

With this statement and following example, we wanted to emphasize the disparity between the general understanding of the importance of factors that rule the search for the vertebrate host and entry into its body with the laconic perception of intrinsic factors that determine parasite location within the host. Therefore, factors important from the epidemiological perspective are not relevant for the navigational choices of the parasite once it entered the body cavities.

It is not clear how the knowledge that was gained from the current study would contribute in a similar manner.

The answer to similar Reviewer’s questions has been discussed in previous points. Nonetheless, we are happy to elaborate more on that topic.

Firstly, lymphatic morphology favors efferent movement of filariae with the mechanism that differs from the physiological enforcement of unidirectional lymph flow by lymphatic valves. That is, the filaria movement through valves is not dependent on pressure differences in the lymphatic segment. Considering the fact that the same morphology but different mechanisms within collectors favor efferent movement of parasites and lymph, it might be possible to target only lymphatic support in filarial efferent migration or design bioengineering devices that move within lymphatic collectors, both without affecting lymph flow.

Secondly, our results exclude a direct dependence of nematode migration on nematode-derived factors, such as metalloproteinases. Despite that, we wondered, how is it possible that a handful of non-conclusive studies showing, for example, a correlative expression of metalloproteinases in filariae, became a foundation for the axiomatic hypothesis where metalloproteinases play a critical role in the lymphatic invasion? In fact, the lack of basic research and the acceptance of wrong assumptions might lead research in the wrong direction and delay the discovery of a treatment that could stop filarial infection.

Lastly, we believe that understanding the mechanism must precede pharmacological intervention as the times when drugs, such as aspirin, were introduced into the market decades before their mechanisms of action were discovered, belong to the past.

We have added the above discussion on the metalloproteinases hypothesis to the modified Discussion.

2. Lack of experimental details – this was an issue throughout the manuscript. The most egregious example of this are the very first figures (Figure 1A and 1B). There are absolutely no details in the method section for these experiments. How were the larvae counted in the pleural cavity? At what time point after injection were they counted? How were the surgeries performed? How soon after surgery were the larvae injected? Why were there only 4 mice in the surgery induced lymphedema group, while there were 8 in the sham surgery group? Were any statistical analyses performed?

Lack of experimental details – this was an issue throughout the manuscript.

We apologize for the lack of technical details and thank the Reviewer for pointing that out. In the revised version of the manuscript, we have added the missing lymphedema and statistical analysis sections. In addition, we explain in detail our modification to the lymphedema model. As a support for the Method section, we have also added Supplementary Technical Figure 1, Technical Figure 2 and experimental schematics to Supplementary Figure 1-3.

Why were there only 4 mice in the surgery induced lymphedema group, while there were 8 in the sham surgery group?

Because of a high number of mice that has to be euthanized before they develop secondary lymphedema, this model is considered as one of the most demanding assays in the lymphatic field (see modified Discussion and our review¹ where we explain the historical problems with animal models of lymphedema). We have optimized many aspects of this method, which increased the success rate from an initial 5-10% to approximately 50% in the presented experiment. Nevertheless, the necessity of removing mice from the experiment according to strict animal welfare rules resulted in a significantly lower number of experimental animals in the lymphedema group. We also performed correct ANOVA and post-priori testing. The test results are marked in the Results section of the manuscript.

The Chy3 model of primary lymphedema that we also used is characterized by low penetrance of the phenotype (in our SPF facility it was 9 mice out of 370 bred, approximately 2% penetrance). However, this model is unique as it produces primary and persistent lymphedema with the occlusion of a collecting lymphatic vessel, leaving the afferent capillary and collecting lymphatics organized in the drainage system as it is in wild-type mice. By comparison, the alternative, a K14-VEGFR3 model is characterized by full penetrance with a complete absence of lymphatics in the dermis, yet the animals do not develop subcutaneous swelling. Therefore, we suspected that in the absence of tissue voids, normally secluded by lymphatic endothelium, the tensile properties of the dermis might change¹, which would favor a compensatory mechanism of filarial dissipation. Specific occlusion of only collecting lymphatics and minimal change in the architecture of the dermis where larvae migrate were the main objectives that determined the choice of these two models.

We have added the above discussion in the Result or Supplementary Methods and Discussion sections.

3. Many claims are made from anecdotal or insufficient evidence – the velocity measurements that were generated for larvae movement through the interstitial space and within the lymphatic vessels appear to be have been made from only one video in each situation. An in-depth analysis of the lymphatic wall breach was performed based on n=1 experiments, as well as the interpretation of the activity of the one larva that was captured entering a collecting lymphatic vessel.

A. Many claims are made from anecdotal or insufficient evidence

We appreciate the Reviewer's comment, but we would also welcome a stricter distinction between our results and conclusions and the anecdotal evidence presented in the literature. We indicated in our Introduction that many aspects of the infection are, in fact, anecdotal with many claims that cannot be tracked down to the original publication. For example, the in vitro secretion of a minute amount of metalloproteinases by infectious larvae of *B. malayi*² led to the generalized presumption that "elastases and collagenases are instrumental in completed filaria infection"³⁻⁵. To strengthen our point, we have added the above example to the modified Discussion.

B. – the velocity measurements that were generated for larvae movement through the interstitial space and within the lymphatic vessels appear to be have been made from only one video in each situation. An in-depth analysis of the lymphatic wall breach was performed based on n=1 experiments, as well as the interpretation of the activity of the one larva that was captured entering a collecting lymphatic vessel.

The primary purpose of the intravital imaging was to provide qualitative clues about how the parasite moves in the tissue. Prior to our imaging, and due to the lack of research, any hypothesis was equally attractive. With this initial direction, we could search for the traces left by migrating filarial larvae, using fixed-tissue imaging techniques. For example, as correctly pointed out by the Reviewer, the anecdotal evidence suggested that filariae digest the tissue with metalloproteinases, which helps them to enter lymphatics. The imaging suggested otherwise. Instead of continuous movement through the tissue, as would be expected if the in-tissue migration relied on the degradation of extracellular matrix outside basement membranes, larvae (as well as saprophytic nematodes) spent most of their time in the tissue on movement without net advancement. These lag periods of migratory inactivity were interleaved with sudden and rapid bursts, during which larvae (or nematodes) could change or reverse the direction of the migration.

This was clarified throughout the manuscript. For example, we added a sentence in the discussion: *“For example, tracking the site of entry of the filaria migrating within a lymphatics revealed that the entry process could heavily damage the wall of the collector. This observation focused our attention on lymphatic injury sites located proximal to the nematode, which otherwise would likely have been viewed as preparation artifacts.”*

However, we could not agree more with the Reviewer; it would be beneficial to collect many more similar videos that would allow calculation of descriptive and inferential statistics for nematodes migrating within the lymphatics. In the revised manuscript, we calculated the average velocities of live filarial movement within the dermis. Together with the in-lymphatic example of filaria (and now also a single free-living nematode) migration, these are the first, known to us, in vivo observations of active migration of any multicellular parasite during its infection phase within the mammal tissues. We matched these two individual in-lymphatic velocities with filaria larvae moving within channels of the maze microfluidic chamber. With these videos, we could exclude numerous potential mechanisms of larval migration, such as passive flow with the lymph in lymphatics, digestion of interstitium (in the absence of movement) in the skin, or fast interstitial movement towards lymphatics, to name a few.

Unfortunately, intravital imaging of filariae/nematodes in the tissue is one of the most demanding methods we have experienced, with a single video obtained once every six experiments. Most probably, if the imaging of the infectious parasites in the mammal tissue was any more straightforward, the infective phase of filaria (or any other helminth) would have been shown at one point during the history of parasitology as a field. As an analogy, the first PCR was not the most efficient method of cloning DNA, but it showed the potential and paved the way for further research.

Nevertheless, we identified three new filariae migration events in the dermis, two of which are shown in the supplementary videos. One is shown in the new Video 6; the second has already been shown in what is now Video 7 (the in tissue movement before entering lymphatics). Tracking of these larvae allowed us to calculate the average velocities for in dermis migration during their migratory bursts.

The directional migration of filarial larvae was inferred from the position of the head and tail from static images. However, dynamic statements of filarial motion were made at this point – “filarial tail to push off against the vessel wall, accelerating larval downstream movement.”

We have added video showing filarial passage through the collector, without the red, larva channel. This video shows that the larvae deform the collector during their passage through the collector by “push off against the vessel wall.” We corrected the mentioned sentence to: “*a larger area of post-valve chamber sinuses (Fig. 1G) provides resistance for the filarial tail to push off against the vessel wall, during the downstream movement of a larva.*”

An in-depth analysis of the lymphatic wall breach was performed based on n=1 experiments, as well as the interpretation of the activity of the one larva that was captured entering a collecting lymphatic vessel.

We were not able to multiplicates results with in-lymphatic filaria migration, but we calculated the average speed of filariae in the dermis during their burst migration (based on three different larvae migrating in the tissue). We have also added essential experiments showing that even without any host adaptations, free-living nematodes were able to forcefully enter lymphatic collectors (Fig. 5 and Supplementary Fig. 5, Videos 11-14), where Videos 14 A and B show the probable in-lymphatic path of the free-living nematode.

The comparison of velocities between lymphatics and interstitium aimed to differentiate between relatively slow tissue migration and fast intra-lymphatic movement. It was confirmed by other indirect data, such as whole-mount imaging of fixed tissue. Nevertheless, we added one more video showing the end of filarial movement within the interstitium. We also point to the video already presented to the Reviewer in which lymphatic entry is preceded by the in-tissue migration. These examples confirm the biphasic mode of tissue movement, with slow movement within the interstitium and add to the calculated speed of in-tissue migration (in total four migratory events were used to calculate in the dermis migration of filarial larvae). The single video of filaria migrating within lymphatics was a unique capture of a rare event. Without determining the average in-lymphatic speed of migration, it shows that this migration could be of an order of magnitude faster than in-tissue migration.

Identification and in-depth analysis of a single confirmed breach were supported by whole-mount images of three other filaria present within damaged lymphatics, indicating the mechanical disruption of the basement membrane during the entry process. Similar, the analysis of a valve effect on filarial movement, was supported with the whole-mount staining of fixed tissues. In the revised version of the manuscript, we show that filariae accumulate preferentially at the efferent (sinus) side of valves with a probability of $p=0.009$ (Fisher exact test). We discuss these results in light of recent result by Proulx et al.(2017) who measured the expected reduction in the lymph flow in the anesthetized mice. These authors showed that mouse sedation virtually halts the lymph flow from the limb. We note that the larva migration within lymphatics was imaged only minutes after intradermal inoculation of filariae when the transient increase in the flow of interstitial fluid and lymph flow is caused by the injection-induced edema. Finally, we present new results showing that even saprophytic nematodes, not evolutionarily adapted to animal parasitism such as bacterivores and plant parasites can also forcefully enter and damage lymphatic collector.

Minor points:

1. Why were the experiments not performed in lymphatic reporter line, such as Prox1-GFP mice? This would allow better visualization of the lymphatics without having to resort to interpretation of the morphology from the in vivo whole mount approach using collagen IV.

We are grateful to the Reviewer for giving us the opportunity to discuss the difference between the reporter protein expressed in lymphatic endothelium and the determination of lymphatic collector nature and functional status based on its basement membrane staining. To support the following discussion, we added the new Supplementary Technical Fig. 2 in which we compare the expression of lymphatic Tomato reporter protein with the staining of basement membrane component, collagen IV. Our staining shows that lymphatic endothelium expresses the reporter protein at varying levels within the same segment of the lymphatic collector. In fact, some segments of the lymphatic vessels ended abruptly, with no reporter protein signal in lymphatic cells, whereas basement membrane staining showed the continuity of the lymphatic vasculature. These types of variability in the expression of reporter protein would have falsely suggested that the lymphatic vessel has been damaged. On the other hand, the characteristic morphology of lymphatics and not their expression profile defines their function. As it was apparent 350 years ago to Olaf Rudbeck, the distinction of the unique morphology of collectors is still quite straightforward, and it is quite impossible to confuse other tissue structures with lymphatics relying purely on their morphology. It is important to note that there is not a single protein known to be expressed exclusively by lymphatic endothelium. In addition, all the lymphatic-associated markers have no lymphatic-specific function linked to them¹. Therefore, identification of a secluded cluster of Prox1-Tomato-expressing cells would have limited meaning as opposed to the identification of a vessel segment by basement membrane staining. Even if that segment was disconnected from the rest of the lymphatic network, the morphology-based identification of the lymphatic fragment would have indicated that it was initially part of the lymphatic vessel. We relied on this anatomical principle during the search of the collector entry site and identified such a location despite the absence of an injury-inflicting nematode. This would not be possible when using a reporter protein of lymphatic endothelium with its uneven and unpredictable expression.

From the methodological standpoint, fluorescent proteins irreversibly lose their properties in a hydrophobic environment. Therefore, they are incompatible with the clearing technique that remains the most optimal from the perspective of tissue morphology and autofluorescence: tissue immersion in benzyl alcohol/benzyl benzoate.

2. Why do the authors make several statements explaining why ear skin is not the best tissue for inoculation of larvae (pg. 6 line 133 to pg. 7 line 144) and then proceed to use ear skin for all the remaining in vivo experiments?

We apologize to the Reviewer for the confusion. We clarify this point in the revised Methods and Results sections of the manuscript. The intravital epifluorescence imaging (and to the extent also the whole-mount analysis of the cleared skin) can be only performed in the thin (virtually 'two-dimensional') tissues, such as the thin dermis of the ear skin. Because epifluorescence does not rely on the time consuming optical sectioning of the tissue it allows fast and long-term imaging of antibody labeled epitopes, sequentially in multiple fields of view. These are critical properties of our imaging method, which in contrast to two-photon or confocal microscopy, allow hunting for rare events such as lymphatic entry or inter-lymphatic migration. Also, only our intravital imaging method⁶ that uses low-intensity excitation light is capable of long-term and low-toxic imaging of labeled cells and likely, the entire organisms such as nematodes. Contrary, two-photon microscopy that uses high-power lasers would rapidly quench any non-protein fluorophores⁶ and rapidly kill TRITC-labeled nematodes. Therefore, using a thin skin, such as the ear dorsal dermis, is the prerequisite for the type of imaging we have performed.

However, using the thin ear skin for the inoculation of nematodes has some significant drawbacks. For example, due to the lack of subcutaneous space, only the intradermal injection can be performed

in the mouse ear dermis. However, distribution by injection of large particles, particularly nematodes, cannot be performed intradermally due to the resistance of dense extracellular matrix, that act as sieve allowing only fluid to be injected beyond the limits of the tissue pocket created by the needle. Because of the sieve effect of the dermis, larvae can only be deposited within a pocket pre-formed by an injecting needle. Yet again, due to the unique anatomy of the mouse ear, in which the dermis is bound to the cartilage, the post-needle dermal pocket does not close after the needle is retracted. This leads to the situation in which injected nematodes are entrapped in the pocket made by the injection needle, with only a few being able to escape into the dermis. We pictured this situation in Videos 1, 2, and 12, which show the entrapment of nematodes within the pocket of the collagen gel (Video 2) or dermis (Video 1 and 12). These videos show the entrapment of injected nematodes within the injection channel left by the needle or in vitro gel chamber. From the above, it is clear that the ear is the least optimal nematode inoculation site. However, we did not intend to create the optimal method for inoculation; we merely wanted to adopt our imaging method for visualization of in-tissue migration and lymphatic entry of nematodes. Therefore, even though the ear is the least optimal dermal location for inoculation of nematodes, it is the only location that can be used for the imaging of nematodes migrating in the dermis using our rapid intravital imaging system. Further, this is also despite the artificial nature of the inoculation, as the inoculation mechanism was not a subject of this study.

Since no study before has contributed a similar way to an understanding of the filarial infection phase, we opted for the lesser evil and used the ear dermis, despite it being a less-than-ideal location for inoculation. We have added this discussion to Supplementary methods and discussion.

3. The authors state that lymph node channels impede further movement of the larvae, but if this were true, how did the larvae reach the pleural cavity in the initial experiments?

It is true that we stated that lymph nodes do impede (but do not block) the migration of filariae, which in fact answers the Reviewer's question. The fact that it takes filaria longer to migrate through the lymph node does not mean that the lymph node blocks larvae migration. Nevertheless, we added the citation, which confirms that lymphatic filariae in humans prefer the pre-nodal and nodal lymphatics, which suggest that depending on filaria species, lymph nodes might delay or even block filaria migration.

4. Figure 4C and 4D do not appear to be discussed anywhere in the results.

The text now refers to Figure 4 and details are explained in the description of Figure 4.

5. The interstitial flow mechanism is greatly weakened (as acknowledged by the authors themselves in the discussion) by the conclusion that the larvae entered at the lymphatic collectors, rather than the capillaries.

It is essential to recognize that tissue fluid flow is not limited to the interstitium-lymphatic capillary compartment. In fact, we stated that in the Discussion. Nevertheless, we clarified the text, which now reads: *"Indeed, our in vitro experiments suggest that fluid current can direct movement of filarial larvae. Tissue currents could, for example, steer nematodes away from blood vessels and guide larvae during or after entry into lymphatic collectors. However, this mechanism might not favor guidance toward the lymphatics as filariae preferentially enter lymphatic pre-collectors. These vessels cannot generate directional fluid currents across the luminal surface, and instead, discharge fluid collected by remote capillary lymphatics..... The fluid currents might instead play an auxiliary role in identifying the direction of lymphatic drainage during the process of lymphatic invasion, and support the direction of filarial movement when the larvae have already entered lymphatics."*

Hence, the potential use of tissue currents to identify the direction of migration during lymphatic entry and navigation within collectors or away from the arteries might be a cornerstone for the universal mechanism of in-tissue migration by filariae but possibly other parasites.

6. The discussion mentions that larvae could move forcefully against the lymph current – where was this shown?

We apologize for the confusion; the text was clarified to point to the fact that larvae, can orient themselves against the valve system, which defines the direction of lymph currents.

7. The discussion regarding chemokine gradients and lateral gene transfer is extraneous and does not appear to be related to the presented data.

We agree with the Reviewer, and the discussion regarding the (in fact anecdotal) effect of chemokine gradients on filaria migration has been moved to the Supplementary methods and discussion.

8. The text contains many spelling errors (e.g. “lymphodema”, collagen “sol”, “colecting”, “as a mouse on egg chew”) and other mistakes throughout

Except “collagen sol,” which is a correct description of the sol of collagen that we used to prepare collagen gel (details about collagen sol-gel transition can be found here⁷), we have corrected any remaining spelling mistakes.

Reviewer #3 (Remarks to the Author):

Mechanical cues guide lymphatic migration of filaria Litomosoides sigmodontis during intial phase of infection by Kilarski et.al.,

One of the questions that has been largely been unanswered in the filarial infections is the mechanism(s) of tissue homing of the infective larvae. While several chemotactic factors, nutritional needs etc. have been proposed, they could not be tested efficiently. In this regard using Litomosoides model, provides a good advance in our understanding some of the early events and generate new hypotheses. The use of brute force entry into the collecting vessels is really nice.

1. It is understandable that sometimes it is better to add in pieces of discussion in the results to help the flow and tell a better story. However, it appears that there is a lot more discussion within the results that can be moved or restricted.

We have moved the entire chemokine discussion from the manuscript to the Supplementary methods and discussion.

2. Except for 14 inoculated ears, there is no mention of the number of mice and the background used in the methodology (except for the Chy-3 with C57BL/6).

We apologize to the Reviewer for the missing part of the Method section. We have added the omitted sections, and details concerning the mouse strains and procedures.

3. The demonstration of the collecting vessels is nice. However, as noted by the authors different filariae home to different tissues. Because this model is limited to the very initial phases and not likely dependent on host susceptibility of other filarial parasites, it would have been more informative to have use other filarial larvae that are easily available such as Brugia malayi and Dirofilaria immitis and have different tissue specificities.

Indeed, we aim to study specifically the L3 migratory phase, therefore, for the simplicity we deliberately wanted to avoid additional factors related to the anti-filaria immunity. However, even if we used the mouse model of *Brugia* lymphatic filariasis, the expected lymphatic entry of its L3 larvae would not be able to discriminate between specific adaptations of parasites and our hypothesis, where randomly migrating nematodes enter encountered pre-collectors. The lymphatic entry and migration would not also be dependent on immunity unless we have vaccinated mice against filaria larvae prior to the migratory experiment. Nevertheless, verifying the mechanism of filaria migration and lymphatic entry in other species of filariae, in immunized mice should be an important follow up to our research.

Instead of *Brugia malai*, we performed experiments that show free-living nematodes are able to migrate in the tissue and enter lymphatics. We also show that similar to filariae, these nematodes display a similar pattern of in-dermis migration and are able to damage vessels during the process of lymphatic invasion. These experiments show that the presented mechanism of lymphatic localization and entry is not dependent on parasitic adaptation and it is likely to be universal to all nematodes. Nevertheless, this hypothesis and the effect of additional factors, such as the filaria-specific immune response, must be verified explicitly in animal models of filaria infection, that is, *Dirofilaria* or *Brugia malai*.

4. The understanding in the field is that the L3's are not 'injected' into the host by the vector. Rather they get released onto the skin near the bite wound and find their way into the host through the bite wound. So it is not clear or how helpful the entrapment analogy can be used to explain the egress of the larvae. I wish there was a way to stain the larvae 'in situ' in mites, that can then be used for infecting mice.

We apologize for the confusion. The entrapment experiments were shown because they can explain the limited nematode spread from the injection site in the ear skin. Yet, we must use that location for our imaging experiment (we discuss this in details in the 'Supplementary methods and discussion'). Importantly, our aim was to investigate the mechanism of tissue navigation and lymphatic entry; therefore, we could "tolerate" the artificial mode of filarial deposition in the tissue.

5. Not all L3's that enter the host develop into adults.

The authors show the varying number of L3s in the collecting vessel after injecting ~30 L3s. Is there a quantitative data of the number of L3's successfully in the collecting vessels compared to L3's stuck in tissue(s)?

By plotting the number of recovered larvae in Fig. 1A and 1B we showed that there is significant variability in the number of larvae recovered from the pleural cavity. These results, however, are in agreement with previous reports showing that irrespectively of infection method, that is, a natural method with the *O. bacoti* vector or subcutaneous injection, the lung recovery was at the level of 15-25% (e.g., Karadjian et al.⁸). Separately, in the Result and 'Supplementary methods and discussion' we discuss the peculiarity of the (qualitative) intradermal inoculation of nematodes that might explain their low frequency of dermis entry.

6. Is the inability of the L3's to penetrate the smaller blood vessels related to the vessel architecture?

Since the diameter of capillaries is smaller than the diameter of filarial larvae, it is unlikely that filariae could enter these vessels even if they managed to puncture capillary blood vessels. Also, the wall of comparable in size blood vessels has additional layers of the elastic media and the thick

adventitia, making these vessels stiffer than the environment of the dermis or pre-collectors' wall. We have added the following statement to the Results section:

“The mechanical reasoning might explain filaria preference for lymphatic pre-collectors and not blood vessels. A migrating nematode can generate a force that is equal to the resistance of the matrix it is embedded in. Therefore, skin-migrating nematodes cannot invade the denser and stiffer wall of blood vessels just as filaria larvae overlaid in the medium could not enter the stiffer dermis (Video 4). Finally, as the diameter of capillaries is smaller than the diameter of filarial larvae, it is unlikely that filariae could enter these vessels even if they managed to puncture capillary blood vessels”.

And the paragraph to the Discussion:

” Firstly, the relatively large (15 μm on average) diameter of lymphatic pre-collectors that is capable of enclosing the entire nematode is similar between mammal species. Secondly, pre-collecting lymphatics lack the structural support of tunica adventitia and elastic tunica media, the vessel layers found in comparable in diameter vein and arteries⁹. Therefore, as the ultimate strength of thin-walled pre-collectors is comparable to the strength or compliance of surrounding intervascular tissue, the lymphatic wall might yield to the invading parasite that attacks the vessel from within the interstitium with a force that depends on the supports (strength or compliance) of that interstitium. According to that mechanical principium¹⁰, nematode could not penetrate the wall of comparable in size but stiffer arterioles or venules, moving in from the more compliant environment of the intravascular interstitium.”

>>>>

Minor comments

1. Pg 6 Ln 132: The labeling for 1D-H was stained for collagen. Should it be stained for VE cadherin and collagen?

The labeling of Fig. 1 has been corrected. The label E was used twice, which was likely a source of confusion. In Figure 1, only panel D shows staining for VE-cadherin, which was also clarified (and discussed) in the figure description and Results section.

2. Supplementary Figure 3 to me was not clear. It would be helpful if there was a cartoon to help understand the anatomical locations being represented.

We added the XYZ directionality vectors to clarify lymph node orientation in Supplementary Fig 3.

3. Typographical errors need to be addressed

The errors throughout the text have been corrected.

7. Pg 6 Lns 118-122 – Is this related to finding the orientation of the worms?

Yes, the orientation of the nematode and the directionality of the valve system can tell us if the worm (at the moment of tissue fixation) was oriented towards the lymph node or in the opposite direction, against the valve system.

We added the following statement in the Results section: *“Labeled morphological structures of filarial larvae allowed the identification of their migratory path against the valve system within lymphatic pre-collectors in the still images of fixed-skin preparations.”*

References

1. Kilarski, W.W. Physiological perspective on therapies of lymphatic vessels. *Adv Wound Care (New Rochelle)* **7** (2018).
2. Petralanda, I., Yarzabal, L. & Piessens, W.F. Studies on a filarial antigen with collagenase activity. *Mol Biochem Parasitol* **19** (1986).
3. Scott, A.L. Lymphatic-dwelling filariae. in *Tropical medicine* (ed. Nutman, T.B.) 283 (Imperial College Press, London, 2000).
4. Kazura, J.W. Lymphatic filarial infections: An introduction to the filariae. in *The filaria* (eds. Klei, T.R. & Rajan, T.V.) (Kluwer Academic Publishers, New York, 2002).
5. Mckerrow, J.H., Caffrey, C., Kelly, B., Loke, P. & Sajid, M. Proteases in parasitic diseases. *Annu Rev Pathol* **1** (2006).
6. Kilarski, W.W., *et al.* Intravital immunofluorescence for visualizing the microcirculatory and immune microenvironments in the mouse ear dermis. *PLoS One* **8** (2013).
7. Tran-Ba, K.H., Lee, D.J., Zhu, J., Paeng, K. & Kaufman, L.J. Confocal rheology probes the structure and mechanics of collagen through the sol-gel transition. *Biophys J* **113** (2017).
8. Karadjian, G., *et al.* Migratory phase of *Litomosoides sigmodontis* filarial infective larvae is associated with pathology and transient increase of S100A9 expressing neutrophils in the lung. *PLoS Negl Trop Dis* **11** (2017).
9. Földi, M. & Földi, E. Structural elements of the lymphatic system. in *Földi's textbook of lymphology* (Elsevier GmbH, Munich, 2012).
10. Vegas, M.R. & Martin Del Yerro, J.L. Stiffness, compliance, resilience, and creep deformation: Understanding implant-soft tissue dynamics in the augmented breast: Fundamentals based on materials science. *Aesthetic Plast Surg* **37** (2013).

Reviewers' Comments:

Reviewer #1:

None

Reviewer #2:

Remarks to the Author:

In the revised manuscript by Kilarski et al., the authors have supplemented their original data with additional imaging experiments that have allowed quantifications of average velocities of nematode larvae migration in tissue. They have also demonstrated that the tissue migration mechanisms exhibited by *Letomosoides* are not specific to this species by demonstrating that other species of nematodes (collected from local soil) migrate in a similar manner. While the new data has improved the manuscript and addressed some of the reviewer's concerns, the text is still presented in a lengthy and confusing manner. Thus, it is still difficult for the reader to appreciate what specific advancements have been made with these experiments.

Major comments:

1. It is still very difficult when reading through the results to understand the rationale for, and the choices made while designing, the experiments. For example, in the first section of the results, it is impossible to determine from the first paragraph which lymphedema model was used in the subsequent experiment. There is description of occluding vessels and of two different transgenic models of mice but it is not very clear from the text in the results which one was used and why. For the saprophytic nematodes, why did the authors decide to inject a variety of unknown species that likely would never be found in mammals, rather than another known model organism?
2. Many experimental details are still missing: how was the imaging performed in Supp Fig 1 and how were the larvae counted. How many mice were injected and imaged in this manner? The text appears to indicate that this experiment was performed on only 1 mouse: "lumbar area of a mouse resulted in the efficient escape of larvae from the injection site" However, a percent escape of the larvae was still calculated. Similarly, how many mice were evaluated in the counting of the larvae within draining lymph nodes? It is not very clear from the statement, "We found a total of 15 filarial larvae in four lymph nodes, two of which were collected immediately (eight filariae, 18 minutes between larval inoculation and lymph node excision) and two others (seven larvae) collected 3 hours after inoculation."
3. The authors have indicated why they preferred to use the basement membrane staining in ears after the *L. sigmodontis* injections. This is understandable considering the defects in the Prox1-TdTom expression patterns that appear to be present in their strain. However, even with these defects, wouldn't it have been more instructive to image the saprophytic nematodes in the lymphatic reporter mice? It is difficult to appreciate from the videos (or from the reconstructed confocal images) that these nematodes are truly moving in lymphatic vessels.
4. The authors should revise the following statement in the introduction: "Even the lymphatic requirement of *Litomosoides sigmodontis*, a common rodent model of lymphatic filariasis in humans, has been speculative for the last five decades and until now it could not be experimentally verified." Reference 18 shows quite clearly in a supplemental figure that a lymphatic phase precedes the invasion into the pleural cavity after subcutaneous injection of *L. sigmodontis*. In light of this statements such as "Here, for the first time, we show that a model filaria, *L. sigmodontis* requires functional lymphatic vasculature to reach the pleural cavity, a destination where it establishes patent infection." should also be toned down.

Minor comments:

What do the authors mean by "terminal collecting lymphatics"? I would understand these to be the terminal trunk lymphatics that merge with the venous system. However, the authors appear to be using this term for collecting lymphatics throughout the body e.g. "Terminal lymphatics within appendages"

Why does it appear that the larvae in Video 2 exhibit motion that is in sync with each other? Both larvae are moving aggressively at the beginning and end of the videos. Is this an artifact of the imaging process? Is this the phototoxicity effect that the authors are mentioning? Is this aggressive movement what the authors are referring to in the following statement: "The phototoxicity was addressed by the introduction of imaging pauses that allowed nematodes to recover, as assessed by their movement activity (not shown)."

The extensive description of lymphatic reporters vs. staining does not belong in the main text. Page 10-11: lines 228-242.

There are still many typographical errors in the text: e.g. from Supp Fig 1 legend: "the differences in mouse chow was a result of the chaw preference at different animal facilities. Arrow point to the injection site"

"infrared dye VivoTag 680 XL NHS" - this dye is in the "near-infrared" wavelength range, not infrared.

Reviewer #3:

Remarks to the Author:

It is good to have the details and the background information. This paper has the novelty and the breakthrough in the field of filarial biology. As mentioned earlier, sometimes additional information is needed for the reader to understand the context and the reason(s) for the experiments and observations. But just like how a diamond needs to be cut and polished, the paper needed some trimming to HIGHLIGHT the key points that got lost in the massive discussion (sometimes often repeated throughout). Moreover, as a result of answering the other reviewer's queries, the pile of extraneous information just got bigger. I feel that the detailed extraneous information while useful, should be moved to the supplementary as much as possible. Also, if data cannot be shown, then it does not belong in the results.

Eg. of the large extraneous material that I think can be moved to the supplementary.

1. Results Section: "Filaria L. sigmodontis requires functional collecting lymphatics to complete its migratory route"

- a. The vast majority of this section to me is discussion/supplementary material.
- b. A line or two introducing the models should be sufficient, as they are described in the methods section and in the discussion.
- c. The main focus of this section to me is lns 122-139 (minus 132-135).

2. Results Section: Fluorescently labeled... Intravital imaging

- a. The anatomical description of the filariae is not required here in the results.
- b. Description of ear model – can be moved to discussion/supplementary methods
- c. Likewise, the advantages of techniques can be minimized here and moved to the discussion/supplementary materials

- d. Again, 'data not shown' – move to discussion/supplementary materials.
- 3. Results Section: Multimodal migration lymphatic collectors
 - a. The entire first paragraph of lymphatic vessel morphology, architecture; choice of staining methodology is discussion material.
- 4. Results Section: Free-living Collecting vessels
 - a. I believe the introduction could be much simpler to something like "To test if this phenomenon observed with *L. sigmodontis* larvae is an adaptation of parasitic nematodes, we performed similar experiments with free-living saprophytic nematodes". This would result in moving Lns 414 – 427 to the methods/discussion/supplementary materials.
 - b. Again, 'data not shown' – move to discussion/supplementary materials

The new discussion on the plausible reasons of the tissue specificity (Lns 621-636) is interesting. Coupled with the discussion/statements in the Chy3/VEGFR3 mice (Lns 117-121), would the authors believe that the L3 larvae (of any species) can use compensatory mechanisms and end up in the blood vasculature and are not limited to lymphatic route (the easier way out as shown in this manuscript) or limited by size?

Also in response to reviewer 2, the authors describe the possibility of using two-photon photodynamic therapy to block collecting lymphatics. This is impractical in filarial endemic area. You never know when you get an infected bite..!! Nor is It practical to treat everybody (over 100 millions individuals. Moreover, if all nematodes use the same pathway, it is going to be universal and not limited to filaria only. This brings me to the title - Should it be even more generalized?

Minor:

- 1. Italicize species names (Lns 171; 174; 632)
- 2. Ln 232: Opening bracket missing.
- 3. Lns 602-603 – *B. malayi*
- 4. Ln 631 – molts
- 5. Ln 700

Response to the Reviewers

We apologize the Reviewers for the lengthy and confusing presentation of our results. The Results section of the revised manuscript was re-written so that each sub-section describe a single problem and it is directly followed by the relevant experimental results. Any additional information was moved to the Supplementary Notes 1–18. Extraneous information from Methods and Discussion were moved to, respectively, Supplementary Methods or Supplementary Discussion. When possible, the sentences followed by the statements: ‘Data not shown’ we removed from the ‘Results’ section. However, in two instances we felt that the manuscript could benefit from the presentation of such data. Hence, the ‘not shown’ statement from the Results sentence: *“Injection-induced edema could not force dead filaria into lymphatics in two control ears, each injected with 50 freeze-thaw larvae (not shown).”*, was replaced with the **new Supplementary Figure 7** (“Intradermal injection could not force filariae into the interstitium”) presented now in the Method section: “Because nematodes are too large to be distributed through the interstitium of the dermis with the intradermal injection (Supplementary Fig. 7), ...”. Also, in response to concerns of the Reviewer 2, the statement presented in the ‘Results’: *“The phototoxicity was addressed by the introduction of imaging pauses that allowed nematodes to recover, as assessed by their movement activity (not shown)”* is now supported by the **new Supplementary movie 14** showing the example of the “transient effect of phototoxicity on the movement of nematodes in vivo”: “The phototoxic effect was inferred from the sudden change in the movement of nematodes, and was addressed by the introduction of 5-10 minutes imaging pauses (Supplementary movie 14).”

Further, the Supplementary **Fig. 1 has a new panel B** that shows the results of the second whole-body (5 hours) imaging experiment, where the subdermal spread of larvae was imaged immediately after the inoculation and 5 hours later (mouse was returned to the cage, and for 5 hours it was not anesthetized). We also complemented the Supplementary Fig. 1C (previously, Supplementary Fig. 1B) with a new movie (**new Supplementary movie 1**) that shows in 69 sequential images taken in 3-5 min intervals the dynamics of larvae spread from the inoculation site and their escape from the skin (the last 4 images were taken 20 hours after the inoculation).

We also calculated the effect size for each of the meaningful parametric comparisons. This turned out to be particularly helpful in clarifying the results from the *in vitro* filariae migration. We also calculated average diameter values for lymphatic capillaries and pre-collectors in the mouse ear dermis and the average diameter of *L. sigmodontis* filariae. We used these values to compare the diameters of lymphatics, filaria larvae and soil-derived nematodes that entered the lymphatics.

Below is the detailed point-by-point response to **the Reviewers’ comments**. The text below that is referenced in the manuscript is underlined. The new text in the manuscript file is marked cyan.

The point-by-point response to the Reviewers’ comments

Reviewer #2 (Remarks to the Author):

In the revised manuscript by Kilarski et al., the authors have supplemented their original data with additional imaging experiments that have allowed quantifications of average velocities of nematode larvae migration in tissue. They have also demonstrated that the tissue migration mechanisms exhibited by *Letomosoides* are not specific to this species by demonstrating that other species of nematodes (collected from local soil) migrate in a similar manner. While the new data has improved the manuscript and addressed some of the reviewer's concerns, the text is still presented in a lengthy and confusing manner. Thus, it is still difficult for the reader to appreciate what specific advancements have been made with these experiments.

Major comments:

1. It is still very difficult when reading through the results to understand the rationale for, and the choices made while designing, the experiments. For example, in the first section of the results, it impossible to determine from the first paragraph which lymphedema model was used in the subsequent experiment. There is description of occluding vessels and of two different transgenic models of mice but it is not very clear from the text in the results which one was used and why.

The text was clarified, and nonessential information was cut out or transferred to the Supplementary Notes 1-18, Supplementary methods and Supplementary Discussion, while the redundant information was deleted altogether.

For the saprophytic nematodes, why did the authors decide to inject a variety of unknown species that likely would never be found in mammals, rather than another known model organism?

We have previously answered a similar question asked by the Reviewer 3: “Even if we used the mouse model of *Brugia* (lymphatic filariasis), the expected lymphatic entry of its L3 larvae would not be able to discriminate between specific adaptations of parasites and our hypothesis, where randomly migrating nematodes enter encountered pre-collectors. “

The injection of random species of soil-derived nematodes showed that even without adaptations for parasitism, nematodes are able to migrate in the tissue and enter lymphatics. Aside from their active migratory status, various sizes and developmental stages of the harvested nematodes relieved us from a pre-determination of properties of tested roundworms. If our generalized hypothesis was correct (i.e., parasites do not need parasite-specific adaptation for host dissemination), then at least some of these nematodes could migrate in the tissue and invade lymphatics (which was the case). However, if we used one of the established mouse models of filariases, for example, the mouse model of *Brugia* infection, the expected results would merely duplicate the *L. sigmodontis* experiments as tissue migration and lymphatic entry of *Brugia malayi*, even if accomplished with similar to *L. sigmodontis* kinetics, could not rule out the adaptive parasitic mechanism of the nematode dissemination.

In other words, repeating the same experiments on parasitic nematodes would not provide conclusive arguments, and in the best case scenario, these experiments would only confirm our prior observations on *L. sigmodontis*. Therefore, we purposely wanted to avoid using any known skin-invading parasitic nematodes as they might have carried specific adaptations for lymphatic invasion. Contrary, the inoculation of soil-derived nematodes might be considered as the control infection with “randomized group of parasites naturally depleted of all parasite-specific properties” therefore they comprise an ideal control, which could exclude any hypothetical or even unsuspected parasitic adaptation that would have allowed tissue migration and lymphatic invasion. Please note that a hypothetical experiment with the inoculation of genetically modified filariae that undergone targeted deletion of mechanisms responsible for skin migration and lymphatic invasion would have also represented an experiment with “**species that likely would never be found in mammals.**” In fact, some of these free-living, bacterivorous nematodes do enter human tissue, for example, the growing number of reported cases of meningoencephalomyelitis caused by *Halickephalobus* species (discussed in the manuscript), and *Pelodera strongyloides*, the bacterivore that populates hair follicles of the skin. Even though not a tissue parasite, the bacterivore *Diploscapter coronata* is an example of an adaptive parasite, that might act as the facultative tomato phytoparasites, but also a facultative zooparasite that invades the urinary bladder and achlorhydric stomach of animals and humans^{1,2}. Likely, with the growing awareness, the number of reports describing free-living nematode infections will increase in the future.

Further discussion on that relevant question is now presented in the Supplementary Note 13.

2. Many experimental details are still missing: how was the imaging performed in Supp Fig 1

This information was already present in the manuscript Method section. Specifically:

The labeling was described under the “Nematode labeling” section:

“Alternatively, instead of TRITC, VivoTag 680 XL NHS ester diluted in DMEM/HEPES medium at a final concentration of 1 mg/ml was used to label L3 larvae for whole-body infrared imaging in a Visen imager.”

In “Subcutaneous injection of filariae” we described the injection procedures:

“Filariae were breech-loaded into a U-100 insulin syringe. After reinstalling the plunger was pushed to remove all residual air from the syringe barrel. Despite prior silanization of syringes and needles, approximately 10%-20% of larvae stuck to the syringe surface. /.../ Fifty VivoTag XL 680-stained filaria larvae in 200 µl of Hartmann’s solution were injected subcutaneously at the dorsolateral side of the lumbar region. “

Under “Live imaging of filariae injected in the skin” section we described the whole-body imaging experiment:

“The subcutaneous spread of 50 infrared (VivoTag 680 XL)-labeled filariae from the dorsolateral lumbar location of a mouse was imaged using the 2D capabilities of a Visen Fluorescence Molecular Tomography device. Immediately after the injection, the mouse that was kept under isoflurane anesthesia was placed in the Visen apparatus and imaged for 2 hours at 3-minute intervals. Afterward, the mouse was returned to its cage. To assess the final spread of filariae, the mouse was anesthetized and imaged again 20 hours later. “

However, considering that in the whole-body imaging experiment, filariae were injected subcutaneously, we agree with the Reviewer that this method should be separately described in Methods. Therefore, the above methodology, starting with: “The subcutaneous spread of 50 near-infrared (VivoTag 680 XL)-labeled...” has its own subtitle:

“Live imaging of filariae inoculated subcutaneously”

and how were the larvae counted.

Under this new Method sub-section, we added the following information:

“Before the injection, larvae were manually counted under bench stereomicroscope while they were collected with a pipette and loaded in the syringe.”

and:

“To account for the systemic escape of larvae, individual larvae were numbered and their count subtracted from the total number of successfully inoculated filariae. Larvae that could not be accounted on the consecutive images were considered as ones that escaped the skin into the systemic circulation.”

How many mice were injected and imaged in this manner?

The experiment summarized in our Supplementary Fig. 1, that is, the imaging of the spread of labeled filariae from the injection site was performed twice, but the detailed analysis of dermal spread and subsequent skin escape was performed once.

To reflect that, we added a new panel B in the Supplementary Fig. 1 that shows the results of the second whole-body (5 hours) imaging experiment, where subdermal spread of larvae was imaged immediately after the inoculation and 5 hours later (mouse was returned to the cage, and for 5 hours it was not anesthetized). We also complemented the Supplementary Fig. 1 C (previously, Supplementary Fig. 1 B) with a new movie (new Supplementary movie 1) that shows in 69 sequential images taken in 3-5 min intervals the dynamics of larvae spread from the inoculation site and their escape from the skin (the last 4 images were taken 20 hours after the inoculation). We apologize that this movie was added only now; this image sequence was lost during the move of our lab from EPFL, Lausanne to the University of Chicago and only recently we managed to recover these files.

The text appears to indicate that this experiment was performed on only 1 mouse: "lumbar area of a mouse resulted in the efficient escape of larvae from the injection site"

The experiment was not performed on one (as shown previously) or two animals but twice, each time on 50 animals. This can be appreciated when one considers the objective of the experiment, that was testing the effect of filaria labeling on their behavior *in vivo*. On the contrary, experiment would be performed on two mice (or one as reported in the previous submission) if the purpose of the experiment was to test, for example, the effect of a mouse on filaria labeling or the spread of labeled filaria (the unlikely effect without prior vaccination, already discussed in the manuscript). Therefore, as we look at the spread of 50 filaria larvae in a single mouse (now Supplementary Fig. 1C), the experiment can be described as performed on 50 individuals (39 successfully inoculated, 9 that successfully escaped the skin) but once (twice considering added data).

However, a percent escape of the larvae was still calculated.

Consequently, in this experiment, the observed variable was categorical (binominal) that resulted in two biologically relevant ratios ("a percent escape of the larvae"): the ratio of larvae spreading from the injection site and the ratio of filaria leaving the skin.

Similarly, how many mice were evaluated in the counting of the larvae within draining lymph nodes? It is not very clear from the statement, "We found a total of 15 filarial larvae in four lymph nodes, two of which were collected immediately (eight filariae, 18 minutes between larval inoculation and lymph node excision) and two others (seven larvae) collected 3 hours after inoculation."

We added the following statement in the Method sub-section ("Intradermal inoculation of nematodes in the dorsal skin of the ear"):

"Only one ear was used for inoculation of filaria or nematodes."

Therefore, in the example given by the Reviewer, four lymph nodes were collected from four mice. One draining lymph nodes equals one mouse as in mice, a single (superficial parotid) lymph node drains the entire outer ear.

3. The authors have indicated why they preferred to use the basement membrane staining in ears after the *L. sigmodontis* injections. This is understandable considering the defects in the Prox1-TdTom expression patterns that appear to be present in their strain. However, even with these defects, wouldn't it have been more instructive to image the saprophytic nematodes in the lymphatic reporter mice? It is difficult to appreciate from the videos (or from the reconstructed confocal images) that these nematodes are truly moving in lymphatic vessels.

Labeling of soil-derived nematodes was highly variable, with larger bacterivores heavily labeled at their mouthparts and less strongly at their cuticle and intestine, while smaller nematodes, particularly the actively moving plant parasites, weakly labeled only at their outer cuticle. Using the same labeling detection range (546-594 nm, generally indistinguishable in filter-based fluorescent illumination) for nematodes and lymphatics could have resulted in masking the weakly stained nematodes by the stronger fluorescent signal from lymphatics in through-the-skin imaging, and render the experimental results uncertain. Further, we did not want to use green or blue fluorophores for nematode labeling as these fluorophores are excited with shorter (this more energetic) wavelengths, therefore are more phototoxic for fluorophore-labeled cells. On the other hand, less phototoxic 647, the near-infra-red label is invisible to the experimenter's eye, and for practical reasons, rapid identification of the rare but essential locations had to be performed with the stereo-(3D) abilities of the human eye (fluorescence stereomicroscope's camera captured images are without the Z-depth component).

However, we are aware of the limitations of the through-the-skin imaging. Despite our efforts in identifying the collecting vessels after the *in vivo* imaging in fixed tissues, these results, in contrast to intravital basement membrane labeling, do not provide the absolute certainty that soil-derived nematodes migrated within lymphatics. Unfortunately, we still do not have the ideal methodology to accomplish that task. Nevertheless, these experiments unequivocally show that without parasitic adaptation, soil-derived nematodes were able to migrate in the skin, and similar to filaria infective

larvae adapt discrete shifts in velocity during their skin migration. When compared to the kinetics of filariae migration, these changes in migration velocity are best explained by the assumption that the nematode migrated within lymphatics. Finally, using a complementary method of whole mount skin preparations, we showed that nematodes from various trophic groups ended up within lymphatics and could migrate to the draining lymph node.

4. The authors should revise the following statement in the introduction: "Even the lymphatic requirement of *Litomosoides sigmodontis*, a common rodent model of lymphatic filariasis in humans, has been speculative for the last five decades and until now it could not be experimentally verified." Reference 18 shows quite clearly in a supplemental figure that a lymphatic phase precedes the invasion into the pleural cavity after subcutaneous injection of *L. sigmodontis*. In light of this statements such as "Here, for the first time, we show that a model filaria, *L. sigmodontis* requires functional lymphatic vasculature to reach the pleural cavity, a destination where it establishes patent infection." should also be toned down.

We cannot agree with the Reviewer's assessment of the quoted results. Firstly, the correlative design of the mentioned experiment shown in the Supplementary Figure 1, ("Migratory phase of *Litomosoides sigmodontis* filarial infective larvae is associated with pathology and transient increase of S100A9 expressing neutrophils in the lung" by Karadjian³) cannot unequivocally exclude the potential contribution of blood vessels in delivering filariae to the lungs. Hence, this experiment, as many reports before (now discussed in Supplementary Note 1), presents a correlation that supports the hypothesis where filariae prefer the lymphatic path, but it does not exclude the potential contribution of the alternative blood vessel route. Therefore, as we might agree with the Reviewer that this reference shows "quite clearly" that there is an involvement of lymphatic path in the *L. sigmodontis* larvae migration towards the pleural cavity, it does not univocally proof that lymphatics are the only route. This is in contrast to our experiment shown in Figure 1B, where the block of lymphatic collectors within the tail, manifested with lymphedema of mouse tail, blocked larvae migration to the pleural cavity.

Secondly, the correlation presented in the mentioned Supplementary Figure 1 shows that on the first day post-inoculation larvae could be found in both, the lymph nodes but also in the lungs (that is, blood vessels of the lungs as opposed to the pleural cavity). Moreover, Fig. 1 that shows that L3 filariae appear in the lungs (L3 filariae naturally (mite) infected or s.c. injected) already 2 to 6 hours after inoculation. In contrast, L3 filariae "are observed in the lymphatics hours to days post the infection peaking at d2 and d3 p.i, before reaching the pleural cavity and accumulating there between d4 and d6 p.i.". However, the specific time (the number of hours) after which larvae were found in lymphatics is not defined (with the exception of Supplementary Fig. 1, discussed above). Therefore, this experiment might actually suggest the opposite, that is, the existence of an alternative (non-lymphatic) path by which filaria larvae are delivered rapidly to the pleural cavity before they appeared in lymphatics. Contrary, our experiment proves that only lymphatics but not blood vessels can be used by the migratory *L. sigmodontis* filariae.

We agree, however, that the pleonastic statement: "for the first time" should be removed from the text of the Introduction: "Here, for the first time, we show that a model filaria, *L. sigmodontis* requires functional lymphatic vasculature to reach the pleural cavity, a destination where it establishes patent infection". Nevertheless, in this manuscript, we indeed show for the first time the absolute necessity of lymphatic route in completing the full migration of filaria *L. sigmodontis* larvae, from the skin to the pleural cavity.

The reasons why we needed to show that infective larvae of non-lymphatic filariae use the only the lymphatic route to reach its habitat in the pleural cavity are discussed in the Supplementary Note 2.

Minor comments:

What do the authors mean by "terminal collecting lymphatics"? I would understand these to be the terminal trunk lymphatics that merge with the venous system. However, the authors appear to be using this term for collecting lymphatics throughout the body e.g. "Terminal lymphatics within appendages"

The Reviewer asks about "**terminal collecting lymphatics**," the term that might indeed be mistaken with terminal lymphatic trunks. We agree that it could have been confusing if the sentence (or its context) was constructed the way the Reviewer pointed. However, the Reviewer quote is missing a critical modifier. The full sentence reads the following: "terminal lymphatics within appendages," which should leave no doubt about the location and the nature of these vessels and should help the reader to differentiate lymphatic collectors within appendages from terminal lymphatic vessels such as lymphatic trunks. In fact, the Reviewers quotes the full term in the second sentence of their comment: "**However, the authors appear to be using this term for collecting lymphatics throughout the body e.g. "Terminal lymphatics within appendages"**".

By using the term: "terminal lymphatics within appendages," we wanted to point to lymphatic collectors that are located within appendages, which is a single vessel in limbs and ear, and two vessels in the tail. Unfortunately, there are no better descriptors, aside from the anatomical specific terms, that could explain the reader the location and importance of these specific lymphatic collectors).

Since the lymphatic described with the full term, that is, "terminal lymphatics within appendages," cannot be confused with other lymphatic vessels in the body we must insist that this term should remain in the manuscript as in our opinion it helps with the text clarity.

Why does it appear that the larvae in Video 2 exhibit motion that is in sync with each other? Both larvae are moving aggressively at the beginning and end of the videos. Is this an artifact of the imaging process? Is this the phototoxicity effect that the authors are mentioning? Is this aggressive movement what the authors are referring to in the following statement: "The phototoxicity was addressed by the introduction of imaging pauses that allowed nematodes to recover, as assessed by their movement activity (not shown)."

We apologize for the unclear description of the Video 2 (currently Supplementary movie 3). This time-lapse movie shows bright-field imaging of unstained filaria larvae in the collagen gel *in vitro*, which can be appreciated by the bright background that is usually not present in the fluorescence imaging. Because there are no artificial fluorophores present, there are no concentrated excitation states that would lead to a pronounced phototoxic effect (as compared to fluorescence imaging). We are not sure about the in sync movement of larvae that concerned the Reviewer; larvae of a specific species of nematodes move at similar frequency and amplitude (discussed in the 'Supplementary Discussion'), which might explain the impression of in sync movement.

The description of the method of this experiment has been copied from the description of the Supplementary movie 3 to the new Method section: "In vitro pocket entrapment of filariae."

Is this the phototoxicity effect that the authors are mentioning?

To dispel doubts of the Reviewer about the phototoxic effect on labeled nematodes, we added two new videos (joined in a single Supplementary movie 14) that show the potential transient phototoxic effect on the nematode movement. The halt in movement activity of the imaged nematode was considered a phototoxicity threshold that triggered the pause in the imaging. After 5 minutes, the imaging was resumed with the reduced power of the excitation light source, and continuous imaging showed that the nematode recovered and continued its movement in the skin.

Is this an artifact of the imaging process?

No, it is not an artifact of image processing. The Reviewer must pardon our ignorance, but we cannot envision how the image processing could lead to artifacts such as in sync movement. Nevertheless, the details of image processing and image encoding are described in Supplementary materials (Microscopy equipment and settings) and for individual movies, in the movie description. In case of the video in question, after setting the minimum and maximum brightness of the image sequence (avoid clipping shadows and saturating highlights), individual images were exported (highest

quality Jpeg format), and the movies were encoded in AVI container with the Xvid codes, the standard procedure (at least before the x.264 encoding became a new standard) that significantly decreases the size of the video file. The movies are now encoded in 4:3 ratio with the lossless option of 8-bit X.264 codec. This procedure significantly increased their quality (but also the file size), but as in the previous version of this movie, it had no effect on the movement of filariae in the movie in question or any other movie.

The extensive description of lymphatic reporters vs. staining does not belong in the main text. Page 10-11: lines 228-242.

The discussion concerning lymphatic reporters vs. basement membrane staining has been moved to the Supplementary methods in the Supplementary information section.

There are still many typographical errors in the text: e.g. from Supp Fig 1 legend: "the differences in mouse chow was a result of the chaw preference at different animal facilities. Arrow point to the injection site"

We apologize the reviewer for added new errors. We do our best to correct them all.

"infrared dye VivoTag 680 XL NHS" - this dye is in the "near-infrared" wavelength range, not infrared.

We changed the "infrared" to "near-infrared" as suggested by the Reviewer.

Reviewer #3 (Remarks to the Author):

It is good to have the details and the background information. This paper has the novelty and the breakthrough in the field of filarial biology. As mentioned earlier, sometimes additional information is needed for the reader to understand the context and the reason(s) for the experiments and observations. But just like how a diamond needs to be cut and polished, the paper needed some trimming to HIGHLIGHT the key points that got lost in the massive discussion (sometimes often repeated throughout). Moreover, as a result of answering the other reviewer's queries, the pile of extraneous information just got bigger. I feel that the detailed extraneous information while useful, should be moved to the supplementary as much as possible. Also, if data cannot be shown, then it does not belong in the results.

Eg. of the large extraneous material that I think can be moved to the supplementary.

1. Results Section: "Filaria L. sigmodontis requires functional collecting lymphatics to complete its migratory route"

a. The vast majority of this section to me is discussion/supplementary material.

b. A line or two introducing the models should be sufficient, as they are described in the methods section and in the discussion.

c. The main focus of this section to me is lns 122-139 (minus 132-135).

The entire paragraph of the 'Results' has been re-written, with a focus on the specific aims and the results. Any additional information has been moved to the 'Supplementary methods' in the 'Supplementary information' section.

2. Results Section: Fluorescently labeled.... Intravital imaging

a. The anatomical description of the filariae is not required here in the results.

Additional information has been moved to Supplementary notes 3 to 5.

b. Description of ear model – can be moved to discussion/supplementary methods

c. Likewise, the advantages of techniques can be minimized here and moved to the discussion/supplementary materials

In response to comments 'b' and 'c,' the description of the ear model was moved to the 'Supplementary notes' 4 and 7, and to Supplementary Methods.

d. Again, 'data not shown' – move to discussion/supplementary materials.

Line 280 of the merged manuscript: Instead of Data not shown, we present a new Supplementary Figure 7.

3. Results Section: Multimodal migration lymphatic collectors

a. The entire first paragraph of lymphatic vessel morphology, architecture; choice of staining methodology is discussion material.

We followed each of the Reviewer's comments and the extraneous information had been moved to Supplementary note 4 and 5, and to Supplementary Methods.

4. Results Section: Free-living Collecting vessels

a. I believe the introduction could be much simpler to something like "To test if this phenomenon observed with *L. sigmodontis* larvae is an adaptation of parasitic nematodes, we performed similar experiments with free-living saprophytic nematodes". This would result in moving Lns 414 – 427 to the methods/discussion/supplementary materials.

The paragraph has been shortened according to the Reviewer comment.

b. Again, 'data not shown' – move to discussion/supplementary materials

Line 444 of the merged file: Instead of data not shown the new Supplementary movie 14 has been added.

The new discussion on the plausible reasons of the tissue specificity (Lns 621-636) is interesting. Coupled with the discussion/statements in the *Chy3/VEGFR3* mice (Lns 117-121), would the authors believe that the L3 larvae (of any species) can use compensatory mechanisms and end up in the blood vasculature and are not limited to lymphatic route (the easier way out as shown in this manuscript) or limited by size?

Coupled with the discussion/statements in the *Chy3/VEGFR3* mice (Lns 117-121)

It is difficult to assess the effect of no-lymphatic phenotype on skin strength/compliance without the comparative data. However, the removal of empty tissue spaces (natural voids in the tissue enclosed only by capillary lymphatic endothelium) might increase the resistance of the dermis, which in turn could help nematodes to puncture blood vessels that would be attacked from the dermis that is less compliant than wild-type skin. If so, the results would reflect the mouse model specificity rather than the physiological properties of the skin. Therefore, the most straightforward approach is to test the necessity of lymphatic path by intervening in the continuity of their collectors, which quite literally form the bottleneck of lymphatic drainage route.

would the authors believe that the L3 larvae (of any species) can use compensatory mechanisms and end up in the blood vasculature and are not limited to lymphatic route (the easier way out as shown in this manuscript) or limited by size?

Since we did not observe a filaria or a nematode within blood vessels nor we found any micro-hemorrhage that would sign the injury to a blood vessel, we can only speculate on the possible mechanism that would lead to successful blood vessel invasion. Limited by their shape and body movement, only nematodes with unique apparatus/tail (i.e., sharp head, larger tail) could puncture less compliant than interstitium blood vessels. For example, all plant parasites are equipped with the various type of stylet within their mouth part and suction apparatus that allows them to attack the plant cells that are reinforced by a rigid cell wall. Without the additional device, only anatomical changes to the head of

the infective filaria would have allowed it to cross through the material of lesser compliance than the interstitium. For example, the narrower head could have created an opportunity to puncture more rigid blood vessels as the puncture pressure in the forward movement would be higher than the pressure exerted by the nematode larvae on the interstitium at its back. In addition to examples given in Supplementary Discussion, here we can also discuss the anatomy of *Onchocerca volvulus* microfilaria, which might also explain their specific fate. These are the only larvae of human filaria that remain in the skin leading to microfilaria-dominant skin pathologies. The larva of *O. volvulus* also has characteristic anatomy, with an acellular and blunt-ended head that is wider than the rest of the body of the larva. *O. volvulus* microfilariae reside in the subcutaneous space and only rarely are found in the blood circulation. This is in contrast to *Loa loa*, filaria that produces microfilaria in similar to *O. volvulus* subcutaneous niche. Contrary to *O. volvulus*, microfilariae of *L. loa* are generally present in the blood. Although similar in diameter (8 μm) and length (300 μm), *L. loa* microfilariae have the cellular and sharply-pointed head, which might explain why these microfilariae could penetrate the (possibly lymphatic) vasculature.

However, even though simpler in terms of the absolute path, the entry into the systemic circulation through blood vasculature does not have to be more straightforward from the perspective of the invader. Broken blood vessels lead to immediate, and localized blood clotting and inflammation, and both of these processes are unnecessary threats to the parasite. The selection benefit for blood invasion would have to cancel the advantage of the already present inherent abilities to invade circulation through lymphatics which by continuity connect to blood circulation. As the evolution “operates in the shortest way possible,” the question should regard benefits that would drive the natural selection towards the less stealth (thus more complex) mechanism for the systemic dissemination. Importantly, even though lymph nodes seem to delay the passage of migrating filariae, their immune environment poses a minimal threat to the live nematodes that are protected by the thick acellular cuticle⁴.

Also in response to reviewer 2, the authors describe the possibility of using two-photon photodynamic therapy to block collecting lymphatics. This is impractical in filarial endemic area. You never know when you get an infected bite..!! Nor is it practical to treat everybody (over 100 millions individuals).

Naturally, the treatment of a large population would be an unrealistic task and I rather meant the potential application of a future offshoot technique that would allow a treatment of selected individuals who, for example, could be at risk for developing tropical pulmonary eosinophilia (i.e., individual who is known to be allergic to microfilaria antigens), a rare (0.5% infected) complication of the patent infection. After all, it's always better to have at the disposition an optional technique that does not require an expensive development.

However, I realized that more than being impractical, the statement in question bears an improper ethical weight. With the exception of its final stage, filariases are diseases curable with annual dosing of the orally-administrated and free-of-charge pill of albendazole or ivermectin. Therefore, even when used as a dialectical argument, a suggestion to treat the endemic population with, likely, an expensive technique might be perceived as parasitism on the victims of these diseases. Therefore, I replaced that statement in the Introduction with the alternative argument (one of the arguments used in that discussion with the Reviewer 2): “As a result of this knowledge paucity, the biomechanical factors assisting larval migration within the tissue are not known, which for example, hinders the investigation of the effect of tissue migrating nematodes on the condition of local lymphatics and in consequence, the skin immunity.”

Moreover, if all nematodes use the same pathway, it is going to be universal and not limited to filaria only. This brings me to the title - Should it be even more generalized?

Indeed, that was the intention when we changed the title of the manuscript. If ancestral properties of nematodes allow them to migrate in the skin and invade lymphatics, likely they would play a role in tissue migration of larvae in other diseases (such as dracunculiasis or ancylostomiasis), but possibly also in the migration of related intestinal parasites (for example, trichinosis). However since we do not have experimental data linking all three classes of nematodes, (mainly members of *Dorylaimia* sub-class, which includes *Trichinellida* parasites (e.g., *Trichinella spiralis*), we could only state that this mechanism is likely typical for most zoo-parasitic nematodes, or more specifically, it is shared between nematodes within the *Chromadoria* sub-class⁵. However, considering the minimal requirements necessary for tissue

migration and lymphatic entry (i.e., nematodes must survive and migrate in the skin, while their girth must match the diameter of the lymphatic pre-collectors), we would expect that migratory larvae of all nematode parasites, can migrate in the tissues and spread systemically via lymphatic circulation. We discuss these hypothesis issues in the Supplementary Note 13 and at the end of the Supplementary Discussion.

Minor:

1. Italicize species names (lns 171; 174; 632) 2. Ln 232: Opening bracket missing.

3. Lns 602-603 – *B. malayi*

4. Ln 631 – molts

5. Ln 700

All the above have been italicized.

Reference

1. Gutierrez Y. Rhabditida. In: *Diagnostic pathology of parasitic infections with clinical correlations* (Gutierrez Y). 2nd. Oxford University Press, USA (2000).
2. Athari A, Mahmoudi MR. *Diploscapter coronata* infection in Iran: Report of the first case and review of literature. *Iran J Parasitol* **3**, 42-47 (2008).
3. Karadjian G, et al. Migratory phase of *Litomosoides sigmodontis* filarial infective larvae is associated with pathology and transient increase of S100A9 expressing neutrophils in the lung. *PLoS Negl Trop Dis* **11**, e0005596 (2017).
4. Brusca RC, Moore W, Shuster SM. The *Nematoda*. In: *Invertebrates*. 3rd. Sinauer Associates, Inc. (2016).
5. Blaxter M, Koutsovoulos G. The evolution of parasitism in *Nematoda*. *Parasitology* **142 Suppl 1**, S26-39 (2015).

Reviewers' Comments:

Reviewer #2:

Remarks to the Author:

The manuscript is more streamlined now. I have a couple further comments on the text:

1) The authors should not be allowed to pool data from the filariae (n=9) and soil derived nematodes (n=2) to reach statistical significance with their "sign test" evaluating for a difference in larvae position on efferent vs. afferent side of the valve.

2) I believe the authors mean "Achilles' heel" not "Achilles' feet" in the first paragraph of the discussion.

Also, I would like to state for posterity (since reviewer comments may be published) that I do not agree with the authors interpretation that injecting 50 filariae into one mouse means that the experiment was performed on 50 animals. The quantification presented is a % escape. This should have been calculated for several mice to generate a mean and standard deviation.

I also would like to clarify that I was asking about the potential artifact of the "imaging process", not the "image processing". The authors should take more care to correctly interpret the reviewer comments before responding in such a manner.

Previous title (NCOMMS-17-25472B):

Inherent traits enable infective filariae to spread through structurally predisposed collecting lymphatic vessels

New title:

Inherent biomechanical traits enable infective filariae to disseminate through collecting lymphatic vessels

Witold W. Kilarski^{1,2,Ω,*}, Coralie Martin³, Marco Pisano², Odile Bain^{3†}, Simon Babayan⁴, and Melody A. Swartz^{1,2,5,Ω,*}

Point-by-point response to reviewers' comments

REVIEWERS' COMMENTS:

Reviewer #2 (Remarks to the Author):

The manuscript is more streamlined now. I have a couple of further comments on the text:

1) The authors should not be allowed to pool data from the filariae (n=9), and soil derived nematodes (n=2) from reaching statistical significance with their "sign test" evaluating for a difference in larvae position on efferent vs. afferent side of the valve.

We appreciate the Reviewer for pointing at that problem as this likely required further discussion.

Firstly, we cannot take the credit for "our sign test." The sign test is a non-parametric test first applied by John Arbuthnot in a paper credited for the first use of the inferential statistics¹. The statistic of the sign test follows a binomial distribution, and the test can be considered as a special case of the commonly used Wilcoxon sign rank test where all deviation magnitudes are tied at 0 or 1 values².

Secondly, please note that we have not merely substituted the p-values of the filariae-only test with smaller p-value from the filariae & nematodes test. In fact, we reported both these values in the following clauses of the Results, with the intention to show the trend of the p-values changes after observations from the soil-derived nematodes were added to the filaria group. Indeed, this resulted in a decrease of the p-value of the pooled groups below the 5% threshold. However, we did not claim that we reached, by itself a rather meaningless³, threshold of "statistical significance." Instead, the presentation of both of these test outputs creates a result itself as only when presented together, these p values showed the reduction trend that coincides with the pooling of the observations for filariae and soil nematodes. Below is a detailed explanation of this reasoning.

The $\alpha=5\%$ threshold for type I error is traditionally used to validate the significance of statistical tests by comparing it to calculated p values, the probability of getting results at least as extreme as what was actually observed. As the α level is an arbitrarily chosen value rather than a physical or given constant, one must interpret the statistical test results in their experimental and biological context. In the example questioned by the Reviewer, we found a total of 9 filariae at the sinus side and 2 at the trunk side of the lymphatic valves (out of 23 found within lymphatics, thus 12 filariae have 0 ties). A sign test produced a two-tailed exact probability $p=0.0654$, by convention an indication of the non-significant result of the test. However, it's closeness to the α suggested that either there was no real difference between trunk and sinus locations of filariae (though with large enough n even biologically irrelevant differences could be found) or that the comparison was underpowered, that is, the real difference could not be revealed because the number of observations was too low or the statistical test was too conservative. Our general hypothesis postulated the presence of a universal to nematodes mechanism that directs the migration of filariae within lymphatics. Because this hypothesis tied filariae and nematodes by their behavior within lymphatics, consequently we decided to pool observation of filariae ($n_{\text{filariae}}=23$, $n^+=9$, $n^-=2$, $n_{\text{null}}=12$) and soil nematodes ($n_{\text{nematodes}}=3$: $n^+=2$, $n^-=0$, $n_{\text{null}}=1$) localization at the valves and re-test the probabilities of finding filariae or nematodes at sinus or trunk sides of lymphatic valves by comparing it to H_0 probability of 0.5 (that is, no preference at the valve sides). It is worth noting that although the sign test is the only test that is

applicable in this situation, it has approximately a 60-70% of the power of its parametric alternative, the one sample t-test².

Contrary, we should not have pooled the observations from filariae and soil nematodes if we suspected that filariae or nematodes intralymphatic migration is governed by different mechanisms. Since there were no arguments advocating any alternative mechanism, not pooling the observations from soil nematodes with filaria larvae would have hindered our analysis by a priori withdrawing already acquired results from data analysis.

To clarify that notion we have modified the appropriate Results section of the manuscript (marked in grey in the text of the Results) to:

Out of 11 larvae (out of 23 located within lymphatics) found in the proximity of valves, only two filariae were located at the trunk (inlet) afferent side of the valve (one is shown in Fig. 1d, Right), while 9 filariae were found at the sinus-(outlet) efferent side of the valves (six larvae are shown in Fig. 1 e-f; $p=0.0654$; the two-tailed signed test for 2 vs. 9 filariae). As our hypothesis postulated a presence of a universal to nematodes mechanism that directs the migration of filariae within lymphatics, we pooled the observations of filariae ($n_{\text{filariae}}=23$: $n^+=9$, $n^-=2$, $n_{\text{null}}=12$) and (discussed later) soil-derived nematodes ($n_{\text{nematodes}}=3$: $n^+=2$, $n^-=0$, $n_{\text{null}}=1$). As a result, out of 26 nematodes found within lymphatics, 11 nematodes were found at the sinus and 2 at the trunk side of the lymphatic valves ($p=0.0225$, the two-tailed sign test for 2 vs. 11 nematodes).

2) I believe the authors mean "Achilles' heel" not "Achilles' feet" in the first paragraph of the discussion.

We are grateful to the Reviewer for pointing to this mistake. We corrected the "Achilles' feet" to "Achilles' heel" in the manuscript text.

Also, I would like to state for posterity (since reviewer comments may be published) that I do not agree with the authors interpretation that injecting 50 filariae into one mouse means that the experiment was performed on 50 animals.

Unfortunately, we cannot fully apprehend the criteria that the Reviewer uses to form their opinion concerning an experiment with multiple individual organisms. To clarify our previous stand on this problem we will use an analogy. Hence, in our opinion, a hypothetical experiment where 50 mice are allowed to escape a maze could be described as a single experiment performed on 50 animals. Similar, in the case of filariae, an experiment on 50 individual filariae that are allowed to escape the skin could be described as a single experiment on 50 animals. We reason that because filaria larvae are independent animals that do not form colonies, herds, nor they are known for displaying any interacting behaviors. Therefore, filariae cannot form a synergistically acting supraorganism (such as, ants or bees) or participate in herd-like coordinated migration, the situations that would justify considering them as one entity. Possibly, the Reviewer might refer to a swarming behavior of nematodes (Supplementary Movie 12); this, however, is an effect of gravity-enforced nematode accumulation in a restricted volume rather than coordinated action of individual nematodes.

The quantification presented is a % escape. This should have been calculated for several mice to generate a mean and standard deviation.

We understand the concern of the Reviewer. As this might create a false impression that these ratios are derived from multiple experiments, we decided to remove the percentage in parenthesis, leaving only the numbers observed.

Nevertheless, the purpose of this experiment was to show that filaria, despite their chemical labeling, can migrate and escape from the live tissue. We test the behavior of filariae by delivering them to subcutaneous space at the back of the mouse because this is the acceptable location for the experimental inoculation of filariae. These experiments likely served its purpose as none of the Reviewers questioned the ability of labeled filaria to migrate within the tissue. However, to further prove that chemical labeling had no immediate effect on larvae behavior, we decided now to replace previously shown two static images (0 h and 5 h) in Supplementary Fig. 1b with the short time-lapse sequence, where subcutaneously injected near-infra-red labeled filariae were first allowed to spread for 18 hours (far beyond our experimental time frame for imaging of intradermally (ear) inoculated filariae). The following time-lapse imaging showed

that at least some filariae that were still present in the skin were alive, with two larvae actively burrowing through the dermis and four others wiggling in the same location (as inferred from changes in their fluorescent intensity). This further confirms that the labeling had no immediate adverse effect on filariae motility *in vivo*.

I also would like to clarify that I was asking about the potential artifact of the "imaging process", not the "image processing". The authors should take more care to correctly interpret the reviewer comments before responding in such a manner.

Clearly, I have wrongly answered the Reviewer question, and I am sorry about that. Below I will try to answer the original comment of the Reviewer. This will hopefully also explain why this mistake was made in the first place.

From the original Reviewer comment:

Why does it appear that the larvae in Video 2 exhibit motion that is in sync with each other? Both larvae are moving aggressively at the beginning and end of the videos. Is this an artifact of the imaging process? Is this the phototoxicity effect that the authors are mentioning? Is this aggressive movement what the authors are referring to in the following statement: "The phototoxicity was addressed by the introduction of imaging pauses that allowed nematodes to recover, as assessed by their movement activity (not shown)."

This comment referred to the Video 2 (now Supplementary Movie 3). Reviewer now highlights the following question : **"Is this an artifact of the imaging process?"**. This question was previously answered by incorrectly assuming that the Reviewer was asking about the description of the video processing technique:

"No, it is not an artifact of image processing. We are not sure how the image processing could lead to artifacts such as in sync movement. The details of image processing and image encoding are described in Supplementary materials (Microscopy equipment and settings) and for individual movies, in the movie description...."

Hence, the Reviewer question referred to "artifacts in the imaging process" of the *in vitro* experiment (filaria escaping the *in vitro* polymerized collagen gel). Artifacts of an *in vivo* process can be defined as experiment-induced departures in biological performance from its original status. However, every *in vitro* experiment by definition is a simplistic, and by this, artificial representations of selected components of the *in vivo* (black box) situation. Therefore, it is difficult to discuss the artifacts of by itself, an artificial *in vitro* experiment. Because I was not expecting such a fundamental question, I must have misinterpreted its original meaning.

To answer the Reviewer question, and considering the above reasoning, this *in vitro* experiment is artificial as the entrapment in the polymerizing collagen type I gel is an artificial process that could never be found *in vivo*. However, this experiment was designed to verify the hypothesis based on prior *in vivo* observations, that is, that fluid-filled non-resistive space within the tissue entraps nematodes. In that sense, the type of entrapping matrix is not relevant to conclusions that could be drawn from the output of the experiment. This experiment reproduced successfully the *in vivo* effect of what was observed after intradermal injection of filaria or soil-derived nematodes. Actually, this is possible an ideal example of the application of the *in vitro* approach in testing the biologically-relevant hypothesis as only in artificial *in vitro* setting we could create proper conditions (by temporarily paralyzing the worm movement during gel polymerization), which allowed a complete enclosure of the filaria within the polymerizing collagen sol. In consequence, the effect of the detainment of filariae within the skin spaces created by the injecting needle was not considered a part of the physiology of the filariae migratory process but as an unavoidable adverse consequence of filaria inoculation by injection.

Why does it appear that the larvae in Video 2 exhibit motion that is in sync with each other?

We cannot confirm the Reviewer observation that filariae from the Supplementary movie 3 (the old Video 2) move in a coordinated (in synch) fashion. However, a simultaneous (though not coordinated) movement might be observed in nematodes that belong to the same species and are at the same developmental stage. This is, however, a result of the shared species-dependent movement characteristic and not a consequence of movement coordination between nematodes (discussed below).

As we stated in the Supplementary Discussion, due to the lack of appendages and circular muscles, nematodes have little flexibility for adaptive modifications. This anatomical simplicity restricts their movements to two main types: head searching and backward-directed waves. The frequency and the amplitude of the movement waves depend on the environment (resistance) in which the larvae migrate. In consequence, infective larvae of a specific species of nematodes (or in fact, all nematodes) switched their movement characteristics between environments in a predictable way⁴. If we compare the two filaria larvae from Supplementary movie 3 (old Video 2), we could notice that the frequency of the larva moving within the gel void is much lower than the other larva that is moving through the gel. An inverse correlation exists between the amplitudes of their movements. This can be explained by the Wallace model of nematode movement^{5,6}, where frequency and amplitude are characteristic for the nematode species, stage and the environment it moves in. directly depend on the number of muscle fibers along with the nematode and within its head and the tail. Hence, as the nematode must increase the force required to move within a denser environment, it modifies the frequency and amplitude of its wave movement but keeps the power output unchanged. In addition, infective larvae of parasites are known to exhibit phases of active-migration and migratory lags after a stimulus. Also, filariae exhibit a type of simultaneous responses: *“Infective larvae of some actively penetrating nematodes may ex-sheathe spontaneously, and use their sheath for support. In this behaviour they extend out of the sheath and wave from side to side, supported only by the posterior few microns. When suddenly stimulated they rapidly reverse into their sheaths in a highly co-ordinated fashion”*⁴. However, rather than coordinated responses, these movements result from the automated reflex to the simultaneously received stimuli. Surprisingly, we observed this behavior in large soil-derived nematodes but not in larvae of plant parasites (Supplementary movie 12, plant-parasitic nematodes are the rapidly moving weakly stained nematodes that resist clustering into the central swarm) or *L. sigmodontis* (Supplementary movie 2, filariae in the post-injection tissue void; through the skin imaging). A changed direction or intensity of fluid flow could act as a stimulus *in vivo*, while in our *in vitro* experiments, the stimuli could result from, for example, the sheer effect of fluid movement (however, the effect should be shielded by the presence of the gel).

This extended discussion is now included in the Supplementary discussion (marked in grey).

Finally, it is unlikely that filariae are affected by the imaging process itself. There is no fluorophore phototoxicity as this time-lapse movie shows bright-field imaging of unstained filaria larvae and filariae, like most parasites, do not have photoreceptors, therefore they do not respond to nonthermal light of visible spectrum. Contrary, soil-derived nematodes might respond to light in a physiological manner, which might explain, together with phototoxicity, the effect of the light on the soil-derived nematode migration and the need for the introduction of imaging breaks in our experiments. This, however, does not affect the relevance of the observation, that is, the ability of soil-derived nematodes to move within the dermis at the multiphase fashion, as the soil-derived nematodes also exhibit migratory lags while moving without the net distance advances (Supplementary videos 13-15).

References

1. Arbuthnott J. An argument for divine providence, taken from the constant regularity observ'd in the births of both sexes. *Phil. Trans. Roy. Soc.* **27**, 186-190 (1710).
2. Sprent P, Smeeton NC. *Applied nonparametric statistical methods*. 4th. CRC Press, Taylor & Francis Group (2007).
3. Nuzzo R. Scientific method: Statistical errors. *Nature* **506**, 150-152 (2014).
4. Croll NA. Behavioural analysis of nematode movement. *Adv. Parasitol.* **13**, 71-122 (1975).
5. Wallace HR. Wave formation by infective Larvae of the plant parasitic nematode *Meloidogyne Javanica*. *Nematologica* **15**, 65-75 (1969).

6. Izquierdo EJ, Beer RD. From head to tail: A neuromechanical model of forward locomotion in *Caenorhabditis elegans*. *Philos. Trans. R. Soc. Lond., B, Biol. Sci.* **373**, (2018).